



**Water yield following forest−grass−forest transitions**
*Running head:* Species changes affect water yield over time
Katherine J. Elliott[1], Peter V. Caldwell[1], Steven T. Brantley[2], Chelcy F. Miniat[1], James M.
Vose[3], and Wayne T. Swank[1]
[1]USDA Forest Service, Southern Research Station, Coweeta Hydrologic Lab, Otto, NC 28763
[2]Joseph W. Jones Ecological Research Center, Ichauway, Newton, GA 31770
[3]UDSA Forest Service, Southern Research Station, Center for Integrated Forest Science,
Raleigh, NC 27695
Correspondence to:  Katherine Elliott (kelliott@fs.fed.us)
Author contributions: WTS and KJE conceived the study and designed the experiment. KJE and
PVC wrote the paper. KJE, PVC, CFM and STB performed the analyses. WTS, JMV, CFM, and
KJE performed the research. KJE, PVC, STB, CFM, JMV, and WTS contributed to discussions
and editing.
*Corresponding author; e-mail: kelliott@fs.fed.us



**Abstract**.  Many currently forested areas in the southern Appalachians were harvested in the
early 1900s and cleared for agriculture or pasture, but have since been abandoned and reverted to
forest (old-field succession). Land use and land cover changes such as these may have altered the
timing and quantity of water yield ($Q$). We examined 80 years of streamflow and vegetation data
in an experimental watershed that underwent forest-grass-forest conversion (i.e., old-field
succession treatment). We hypothesized that changes in forest species composition and water use
would largely explain long-term changes in $Q$. Aboveground biomass was comparable among
watersheds before the treatment (208.3 Mg ha$^{-1}$), and again after 45 years of forest regeneration
(217.9 Mg ha$^{-1}$). However, management practices in the treatment watershed altered resulting
species composition compared to the reference watershed. Evapotranspiration (ET) and $Q$ in the
treatment watershed recovered to pretreatment levels after nine years of abandonment, then $Q$
became less (averaging 5.4% less) and ET more (averaging 3.4% more) than expected after the
10$^{th}$ year up through present day. We demonstrate that the decline in $Q$ and corresponding
increase in ET could be explained by the shift in major forest species from predominantly
*Quercus* and *Carya* before treatment to predominantly *Liriodendron* and *Acer* through old-field
succession. The annual change in $Q$ can be attributed to changes in seasonal $Q$. The greatest
management effect on monthly $Q$ occurred during the wettest (i.e., above median $Q$) growing
season months when $Q$ was significantly lower than expected. In the dormant season, monthly $Q$
was higher than expected during the wettest months.
**Keywords**: diffuse-porous, evapotranspiration, forest succession, paired watersheds, ring-
porous, water yield



## 1 Introduction

Forests play a critical role in regulating hydrological processes in headwater catchments by

moderating the timing and magnitude of streamflow (Burt and Swank, 2002; Chang, 2003; Ice

and Stednick, 2004; Ford et al., 2011b; Vose et al., 2011). Hydrological processes in forests are

particularly sensitive to disturbances that reduce tree vigor or leaf area and thus decrease

evapotranspiration (ET) (Aranda et al., 2012; Edburg et al., 2012; Brantley et al., 2013). Most

efforts at studying the effects of disturbance on watershed hydrology have focused on

quantifying the effects of forest harvesting practices on water yield ($Q$) (Bosch and Hewlett,

1982; Stednick, 1996; Burton, 1997; Brown et al., 2005; Wei and Zhang, 2010; Ford et al.,

2011a; Zhang and Wei, 2012; Liu et al., 2015). Reviews have shown that, in general, harvesting

<20 % of the basal area shows no detectable increase in annual $Q$; but, $Q$ increases thereafter as

the percentage of basal area harvested increases (Bosch and Hewlett, 1982; Andréassian, 2004;

Brown et al., 2005). However, recent work that aims to merge ecology of the resulting forest and

species composition with traditional hydrology approaches (i.e., ecohydrology) has advanced our

understanding greatly. For example, the Brantley et al. (2013, 2015) showed that lasting changes

in annual $Q$ (lower) and persistent changes in peakflow (20%+, after the most intense storms)

were observed with only about a 5 % basal area loss of eastern hemlock (*Tsuga canadensis* (L.)

Carrière),

Most of the Eastern U.S. forests have been harvested at least once since the late 1800s

(Yarnell, 1998; Foster et al., 2003; Thompson et al., 2013; Martinuzzi et al., 2015); and many

forested areas have undergone forest to agriculture land use changes, and then been abandoned to

revert back to forest (i.e., abandoned agriculture or old-field succession) (Otto, 1983; Trimble et





al., 1987; Wear and Bolstad, 1998; Bellemare et al., 2002; Alvarez, 2007; Thiemann et al., 2009;
Ramankutty et al., 2010; Kirk et al., 2012). Land abandonment has also been prevalent and
ongoing since the early 20th century in other countries (Cramer et al., 2008; García-Ruiz and
Lana-Renault, 2011). Land use and land cover (LULC) changes, such as forest–grass–forest
transitions, may alter the timing and quantity of $Q$. Because land use conversion from forests to
agriculture often includes a combination of changes in vegetation composition and soil physical
attributes, it is difficult to separate the effects of vegetation changes from soil changes (see
reviews by Neary et al., 2009; Zimmermann et al., 2010; Houlbrooke and Laurenson, 2013,
Morris and Jackson, 2016). Land cover conversion that requires heavy machinery or includes
livestock grazing decreases soil infiltration and saturated hydraulic conductivity (e.g., Hassler et
al., 2011; Price et al., 2011; Morris and Jackson, 2016), and can thus increase peak flow during
storms, and flood frequency and severity (Reinhart, 1964; Hornbeck, 1973; Burt and Swank,
2002; Alila et al., 2009; Green and Alila, 2012). Without soil compaction and alteration of water
flow pathways, forest trees typically use more water and extract water from deeper soil than
shallower-rooted grasses (Zhang et al., 2001; Kulmatiski and Beard, 2013), which could result in
higher ET and lower $Q$ at the catchment scale.

Several studies have compared $Q$ and ET by forests and pastures. Analyzing 250 catchments

worldwide, Zhang et al. (2001) found that forested catchments had higher ET than grass
pastures, with few exceptions. Replacing trees with grass cover generally increases $Q$ by
decreasing ET (Hibbert, 1969; Bosch and Hewlett, 1982; Farley et al., 2005), although not
always (Brauman et al., 2012; Amatya and Harrison, 2016). In some basins when agricultural
land use is reduced and forest cover increased, $Q$ is unchanged, and can be explained in part by
the species-specific traits in water use (e.g., deciduous vs. evergreen, and/or late season vs. early





season perennial grass) studied (Cruise et al., 2010), and the geomorphological differences
among biomes (Zhou et al., 2015).

Large differences among tree species in their leaf and canopy conductance, transpiration per

unit leaf area, and whole tree water use for any given diameter exist in eastern temperate
deciduous forests  (Wullschleger et al., 2001; Ford and Vose 2007; Ford et al., 2011a). This is
especially true when comparing hardwoods within diffuse-porous and ring-porous xylem
functional groups (Taneda and Sperry, 2008; Ford et al., 2011a; von Allmen et al., 2015).
*Liriodendron tulipifera*, a diffuse-porous species common to the eastern temperate deciduous
forest biome, has among the highest transpiration rates of forest trees; while *Acer rubrum* L. and
*Betula lenta* L., also common diffuse-porous species, rates are lower than *L. tulipifera.* However,
they have relatively high transpiration rates compared to common ring-porous *Quercus* species
(Ford et al., 2011a).

Few studies have examined long-term changes in catchment hydrology through a forest−

grass−forest transition, with specific attention focused on species compositional changes and
their effect on ET and *Q*. A treated watershed within the Coweeta Hydrologic Laboratory,
western North Carolina experienced this LULC transition, and reported similar *Q* between forest
and grass when the grass cover was well fertilized (Hewlett and Hibbert, 1961; Hibbert, 1969;
Bosch and Hewlett, 1982; Burt and Swank, 1992). However, they did not investigate why *Q* was
lower than expected after grass cover abandonment and through the early successional
development of the deciduous forest. Road construction could be a contributing factor because
installing temporary roads to facilitate timber harvesting can affect hydrology (Harr et al., 1975;
Alila et al., 2009), but only 3.3 % of the watershed area was in temporary roads (inactive for the
least 50 years). In addition, roads comprising less than 6 % of the watershed area appear not to





change storm hydrographs significantly (Harr et al., 1975; Swank et al., 2001; Alila et al., 2009).

In a more recent study, Ford et al. (2011b) suggested that the decline in $Q$ over time could be due

to a shift in the dominant tree species in the treated, old-field succession watershed.

Working in the same experimental watershed as authors above, we compared the long-term

changes (1934−2015) in: 1) aboveground biomass, leaf area index (LAI) and species and

functional (xylem anatomy) group composition; 2) estimated growing season mean daily water

use (DWU); 3) annual water-balance derived ET; and 4) daily, monthly, and annual $Q$ between

the treated, old-field succession, watershed (WS6) and nearby reference (WS14, WS18)

watersheds with an emphasis on the period of reforestation. We hypothesized that: 1) a shift in

species composition and their attendant DWU will largely explain long-term changes in $Q$; 2)

annual $Q$ would be lower in the treated WS6 through forest succession concurrent with greater

DWU with additional changes in timing of $Q$ due to altered species composition; and 3) monthly

$Q$ would be greater in the treated WS6 for wet periods (high or peak flows) and this effect would

be greatest in the dormant season.

## 2 Materials and Methods

### 2.1 Study area

The study was conducted at the USDA Forest Service Coweeta Hydrologic Laboratory, a 2185

ha forested basin in the Nantahala National Forest in North Carolina, U.S. (35° 06′ N, 83° 43′

W). Climate in the Coweeta Basin is classified as marine, humid temperate (Swift et al., 1988),

with mild temperature (average 12.8 °C) and ample precipitation (average 1795 mm yr$^{-1}$). Three





watersheds (WS6, WS14 and WS18) within 1 km of one another, similar in elevation, slope, and
aspect were used in this study (Table 1). WS14 and WS18 are untreated reference watersheds.
WS6 experienced a disturbance regime similar to forest conversion to pasture and subsequent
abandonment common across the region (see below). Soils in all watersheds are moderately
permeable, well-drained, moderately deep to very deep, and with a saprolite layer up to 6 m deep
(Thomas, 1996).

**2.2  History of disturbance**

Before 1842, the Coweeta Basin was burned semiannually (Douglass and Hoover, 1988).
Between 1842 and 1900, light semiannual burning and grazing continued. From 1912 to 1923
heavy logging occurred (Douglass and Hoover, 1988). Loss of American chestnut (*Castanea*
*dentata* (Marshall) Borkh.) in the 1930s (Woods and Shanks, 1959; Elliott and Swank, 2008)
was followed by loss of *Tsuga canadensis* (L.) Carrière over the last decade (Elliott and Vose,

2011).

The disturbance regime in WS6, the treated watershed, was extensive (Table S1). In July
1941, 12% of the catchment (1.06 ha area) along the stream was cut to determine how riparian
vegetation affects $Q$ (Dunford and Fletcher, 1947). In 1958, the entire watershed was clear-cut,
merchantable timber was removed, and the residue was piled and burned. In 1959, surface soil
was scarified and seeded to *Festuca octiflora* grass. In 1960, the watershed was treated with a
one-time application of 1100 kg ha$^{-1}$ lime, 110 kg ha$^{-1}$ 30-10-10 NPK and 18.4 kg ha$^{-1}$ granular
60% potash. Between 1960 and 1965, *Kalmia latifolia* L.*, Rhododendron maximum* L.*,* and other
hardwood sprouts were suppressed with spot applications of 2,4D [(2,4dichlorophenoxy) acetic



acid] to maintain the watershed in grass cover (Hibbert, 1969). In 1965, the watershed was
fertilized again, as above. In 1967, the grass was herbicided with atrazine [2-chloro-(4-
ethylamino)-6-9-isopropylamino)-Strizine], paraquat [1,1 dimethyll 4,4 bipyridinium ion
(dichloride salt)], and 2,4D [(2,4dichlorophenoxy) acetic acid] (Douglass et al., 1969), and then
left undisturbed (hereafter, old-field succession). Although the grass was not cut or grazed, the
lime and fertilizer amendments with attendant high productivity and nutrient uptake by the grass,
*Festuca octiflora*, make these applications somewhat similar to agricultural practices. The
original objectives of the conversion from forest-to-grass were to compare water use of grass
versus hardwoods (Hibbert, 1969; Swank and Crossley, 1988) and to determine how conversion
to grass affects discharge characteristics (Burt and Swank, 1992).

Two adjacent forested watersheds (WS14, WS18) were selected as references to provide an

adequate number of sample plots (described below) for analysis of changes in vegetation. These
reference watersheds with similar physiography (Table 1) are characteristic of mature, second-
growth hardwood forests, and have remained unmanaged since 1923 (Swank and Crossley,
1988). We considered forest age for WS14 and WS18 since the loss of *C. dentata* presently to be
75+ years old.

**2.3 Measurements**

**2.3.1 Vegetation**

The relative importance of woody species over time was characterized with repeated tree
surveys. In treated WS6, surveys were conducted in 1934, 1958, 1982, 1995 and 2012. In 1934,





only five 0.08 ha permanent plots were measured along the east-side of the watershed; in all
subsequent surveys, plots were placed across the entire watershed. In 1958, a pretreatment strip
inventory sampled 25 % of the watershed area with 10 m wide strips approximately 40 m apart
extending along transects from the ridge-top to the stream channel. This sampling method
resulted in a total of 37 unequal sized plots (ranging from 0.02 to 0.14 ha), including the riparian
corridor. In 1982, thirty-four 0.02 ha plots were permanently marked continuously along five
transects from ridge-top to near stream; these 34 plots were re-measured in 1995 and 2012.

In reference WS14, thirty-one 0.08 ha permanent plots, were surveyed in 1934, 1969, 1988–

1992 (hereafter, 1992), and 2009–2010 (hereafter, 2009). In reference WS18, eight 0.08 ha
permanent plots were surveyed in 1934, 1953, 1969, 1992, and 2009.

In all watersheds and for all survey periods, diameter of woody stems $\geq$ 2.54 cm at diameter

at breast height (DBH, 1.37 m above ground) was measured by species and recorded into 2.54
cm DBH classes. In 1934, only percent cover was recorded for the two evergreen shrubs,
*Rhododendron maximum* and *Kalmia latifolia*; for this reason, we do not estimate biomass and
leaf are index (LAI, $m^2$ of leaf area $m^{-2}$ ground area) for these species in 1934. In all other years,
stem diameters of these evergreen shrubs were measured in the same manner as the tree
diameters. Median DBH values were used to calculate basal area, aboveground biomass, and
LAI. We used species-specific allometric equations developed on-site to estimate the
aboveground biomass and LAI contribution of each species in each watershed (McGinty, 1972;
Santee and Monk, 1981; Martin et al., 1998; Ford and Vose, 2007; B.D. Kloeppel, unpublished
data; C.F. Miniat, unpublished data). Species nomenclature follows Kirkman et al. (2007).

**2.3.2  Water yield ($Q$) and evapotranspiration (ET)**




We used both chronological-pairing (i.e., corresponding to the same meteorological input) and
frequency-pairing (described below) analyses to detect potential hydrologic responses of $Q$ and
ET to land use and land cover change. Both analyses used the paired watershed approach (Wilm,
1944; Wilm, 1949). The chronological pairing approach allowed us to create a time series of
estimated change in annual $Q$ and ET over the period of record and to relate these changes to
both the treatment and to climate. In addition, this analysis allowed us to determine when a
consistent change in $Q$ began, enabling us to establish the time period of interest for the
frequency pairing. The frequency pairing approach allowed us to compare the post-treatment
distribution of monthly and annual $Q$ to that of the pretreatment period. We used WS18 and WS6
as the reference and treatment watersheds, respectively. We did not compare WS6 to WS14
because there were gaps in the WS14 flow record in the years immediately following the grass
conversion and herbicide application. For both watersheds, 5-min stream stage was used to
estimate $Q$ (Reinhard and Pierce, 1964; Swift et al., 1988).
We modeled WS6 annual $Q$ as a function of WS18, incorporating the effect of grass
conversion and reforestation treatments over time. Annual $Q$ was computed on a May–April
water year to minimize the effects of year-to-year changes in storage, as soils are generally at
their wettest by the beginning of May. Models had the following form:
$$\hat{Q}_T = a + bQ_R + eM1t1 + \left[ M2c \left( h - \frac{1}{1 + exp^{-t2}} \right) \right], \quad (1)$$
where;
$\hat{Q}_T$ = predicted $Q$ from treated watershed WS6 (mm yr$^{-1}$),
$Q_R$ = measured $Q$ from reference watershed WS18 (mm yr$^{-1}$),





M1 = management representing grass conversion; M1 = 1 for water years between and including
1960 and 1966, M1 = 0 otherwise,
t1 = time since grass fertilization; t1 = water year – fertilization year for water years between and
including 1960 and 1966 where fertilization years include water years 1959, 1961, and 1966, t1 =
0 otherwise,
M2 = management representing reforestation after grass conversion; M2 = 1 for water years
greater than or equal to 1967, M2 = 0 otherwise,
t2 = time since reforestation after grass conversion; t2 = water year – 1967 for water years
greater than or equal to 1967, t2 = 0 otherwise,
$P$ = annual precipitation (mm yr$^{-1}$)
$a, b, c, e, h$ are fitted parameters.
The increasing linear $Q$ response after fertilization ($eM1t1$) accounts for the decline in annual
grass production and water use as noted by Hibbert (1969). All models were fit using PROC
NLIN (SAS v9.4, SAS Institute, Cary, NC).
We define the treatment response, $D$, as the difference in $Q$ in the treated watershed $Q$ ($Q_T$)
from that predicted by the reference watershed:
$$\boldsymbol{D} = Q_T - \left( \hat{Q}_T; \ M1, M2 = 0 \right). \qquad (2)$$

The proportion of the variability explained by the model was quantified using the ratios of the
error-to-total sum of squares and the total-to-error degrees of freedom as:
$$R^2_{adjusted} = 1 - \frac{SS_E}{SS_T} \times \frac{df_T}{df_E}. \qquad (3)$$

Parameter estimates were interpreted as statistically significant at $\alpha = 0.05$. Annual ET was
computed as precipitation ($P$) – $Q_T$, assuming the largely impermeable bedrock underlying the
Basin that results in negligible deep groundwater losses (Douglass and Swank, 1972). Watershed





*P* was estimated using a nearby eight inch (20.3 cm) National Weather Service standard rain
gauge, SRG 96 (Laseter et al., 2012).

**2.3.3 Frequency-pairing flow distributions**

We used the frequency-pairing method (Alila et al., 2009; Brantley et al., 2015) to detect
differences in frequency between observed and predicted annual and monthly *Q* after treatment.
Briefly, frequency-pairing is an analytical method that quantifies differences in observed and
predicted *Q* parameters based on the probability of occurrence of a given *Q* (or flow at a given
probability) rather than based on occurrence at a discreet time (i.e., chronological-pairing). This
accounts for rainfall amount and antecedent soil conditions. We used pre-treatment *Q* during
water years 1939–1941, 1948–1951, and 1956–1958, to estimate the expected cumulative
distribution functions (CDFs, $F_Y$) for observed and predicted *Q* in the treatment watershed using
the linear regression equation:
$$\hat{Y}_i = b_0 + b_1 X_i, \qquad (4)$$
where, $X_i$ is the observed *Q* in the reference watershed for period *i* (day of year) and $\hat{Y}_i$ is the
expected *Q* for the treatment watershed under undisturbed conditions for the same period. We
used PROCMODEL (SAS v9.3, SAS Institute, Cary, NC) to predict monthly post-treatment *Q* in
the treatment watershed from May 1979–Apr 2015 and annual post-treatment *Q* for water years
1980–2015. To model monthly *Q*, we separated the data by calendar month and created twelve
separate regression equations. Using separate regression equations for each month helped
account for variations in paired watershed *Q* relationships among months and helped to
distinguish differences in effects among seasons.





Observed and predicted $Q$ values were then plotted as an estimate of the probability of
occurrence for ranked event $Y_{(i)}$ during any time period $i$. The exceedance probability, $1$-$p$, was
estimated for each period using the equation:
$$1 - F_Y\left[\hat{Y}_i\right] = \frac{m - 0.40}{n + 0.20},\qquad(5)$$
where, $m$ was the rank for a given flow and $n$ was the total number of flow periods in the
distribution. This function provided an empirical estimate of the quantile for a given flow value
(Cunnane, 1978; Stedinger et al., 1993). Confidence limits for each predicted flow at each
probability of occurrence were estimated as:
$$Y_m \pm z_{1-\frac{\alpha}{2}}\sqrt{(Var_1\left[Y_m\right] + Var_2\left[Y_m\right])]}.\qquad(6)$$
We used a pair of Monte-Carlo simulations to estimate the variability associated with the
predictive uncertainty in equation ($Var_1$), and the uncertainty associated with the sampling
variability at each rank ($Var_2$). For these analysis, we used 1000 iterations for each simulation.
We used the raw, expected post-treatment values from equation (5) to correct for the loss of
variability in the upper tails of the distribution (Alila et al., 2009). The CDFs were then used to
construct flow duration curves to assess changes in untransformed $Q$ at monthly and annual
intervals by comparing the change in magnitude for a given probability or the change in
probability for a given magnitude (Alila et al., 2009; Green and Alila, 2012).

**2.3.4  Growing season daily water use (DWU)**

Plant water loss was estimated by scaling up sap flux measurements of numerous species and
diameter sizes at Coweeta Hydrologic Laboratory (Ford and Vose, 2007; Ford et al., 2011b;
Brantley, et al. 2013; Miniat, unpublished) using methods outlined in Ford et al. (2011a). We fit



the observed growing season mean daily water use (DWU, kg day$^{-1}$) to stem DBH (cm) using a
power function of the form:

$$DWU = b_0 * DBH^{b1} \qquad (1)$$

Species were grouped into xylem functional types (diffuse-porous, ring-porous, semi-ring
porous, evergreen shrub, or tracheid) and growing season DWU models were developed for each
xylem functional type. For example, *Carya* spp. have semi-ring porous xylem; *Quercus* spp. and
*Oxydendron arboreum* have ring-porous xylem; and *Betula lenta*, *Liriodendron tulipifera*, and
*Acer rubrum* have diffuse-porous xylem (Table S2). Because *R. pseudoacacia* behaves more like
a diffuse-porous species, its measured values of DWU and DBH were combined with the
diffuse-porous model. Even though *Robinia pseudoacacia* has ring-porous xylem, it is isohydric
(i.e., maintaining stable leaf water potentials as soil water potentials drop, Klein, 2014) and has
higher DWU than *Quercus* or *Carya* (Miniat and Hubbard, unpublished). For the two understory
evergreen species, *Kalmia latifolia* and *Rhododendron maximum*, we applied the mean DWU
value from 16 instrumented shrubs because DWU models based on DBH alone provided limited
predictive power (Table 2). We estimated growing season mean plot DWU by modeling DWU
by functional type and vegetation surveys by diameter for all watersheds. We did not estimate
DWU for the 1934 survey, when *C. dentata* was most abundant, because most of the trees had
been affected by chestnut blight compromising their functional xylem.

**3  Results**

**3.1  Vegetation dynamics**





Prior to treatment, species composition and aboveground biomass among the watershed were
similar (Fig.1). In 1934, aboveground biomass was comparable among the treated WS6 and
references WS14 and WS18 averaging 200 Mg ha$^{-1}$ ($p = 0.706$) (Fig. 1a, Table S3). Biomass
declined in WS6 (99.51 Mg ha$^{-1}$) from 1934 to 1958 prior to conversion to grass, and in WS18
(148.42 Mg ha$^{-1}$) from 1934 to 1953 (Table S3). The decline in biomass and LAI between 1934
and the 1950s was primarily due to the loss of *Castanea dentata* (Fig. 1a–c). In 1934, *C. dentata*
occupied from 40–54 % of the biomass and 29–43 % of the LAI across the three watersheds
(Fig. S1).
The grass cover in the treated watershed was highly productive, but following the herbicide
treatment (i.e., old-field succession), early-successional vegetation rapidly established (Fig. S1a).
During the five years when WS6 was maintained in grass, biomass ranged from 5.67 to 7.30 Mg
ha$^{-1}$. In 1968, one year after cessation of treatment, the aboveground biomass was 3.92 Mg ha$^{-1}$ in
WS6. At that time, the one year old field was dominated by *Erechitites hieracifolia* (L.) Raf.,
*Phytolacca americana* L., *Eupatorium* spp., *Equisetum arvense* L. and had remnants of *Festuca*
*octiflora*. In the years between 1968 and 1982, WS6 was rapidly colonized by *Robinia*
*pseuodoacacia* and *Liriodendron tulipifera* (Fig. S1a); whereas the most abundant species in the
reference watersheds in the years following the loss of *C. dentata* (1969 to 2010s) were *Quercus*
spp. and *Acer rubrum* (Fig. S1b–c; Tables S4–S6).
Forest composition following grass cover was biased towards tree species with deep
functional sapwood and diffuse-porous xylem. In 1934, all watersheds were dominated by
species with semi ring-porous (*C. dentata* and *Carya*) or ring-porous (*Quercus*) xylem,
accounting for more than 80 % of the aboveground biomass (Fig. 2a–c) and 80 % of the LAI
(Tables S4–S6). Although species with semi ring-porous xylem declined in all watersheds over




time, the increase in species with diffuse-porous xylem was greater in the treated watershed
compared to reference watersheds (Fig. 2a–c). As the young forest developed following grass
herbicide and abandonment, species with diffuse-porous xylem and *R. pseuodoacacia* dominated
forest biomass, while species with ring-porous xylem were only 2.7 %. By 2012, 93 % of
vegetation in the treatment watershed was comprised of species with diffuse-porous xylem (Fig
2a), while the reference watersheds were about half of the species with ring-porous xylem (Fig.
2b–c).

**3.2  Water yield (*Q*) and evapotranspiration (ET)**

The forest−grass−forest treatment of WS6 resulted in significant effects on *Q* over time. Models
of annual *Q* explained more than 98% of the variability in *Q* over the period of record. Initial
harvesting increased *Q* by 99 mm (10.5 % above the expected *Q*) in 1960 (Fig. 3), and *Q*
remained higher than expected during the grass conversion period except in 1959, 1961, and
1966 when grass production was highest due to fertilizer application. The largest treatment effect
occurred in 1967 when herbicide was applied to the watershed, resulting in a *Q* increase of 259
mm (31 % above the expected *Q*) (Fig. 3). *Q* remained higher than expected for approximately
nine years after the herbicide treatment as the vegetation re-established. Beginning in 1977 and
continuing through 2015, *Q* was less than expected in 32 of 35 years (Fig. 3), suggesting that the
new forest used more water (i.e., had higher ET) than expected had it not undergone treatment.
Since 1980, on average, annual *Q* decreased by 6.1 %, ranging from a *Q* increase in 1981 of 30
mm (+5.5%) to a decrease of 142 mm (16%) in 2003. ET (not shown) increased by 4.5 % on
average relative to what was expected in the absence of management.




### 3.3 Changes in flow distribution

In addition to the forest-grass-forest treatment changing the amount of Q, it also fundamentally changed the distribution of Q, with the most pronounced changes at the height of the growing and dormant seasons. The annual and monthly $Q$ relationships between the reference and treatment watersheds for the pre-treatment period were highly significant (annual, $n = 10$, $r^2 = 0.97$, $p < 0.001$; monthly, $n = 10$, $r^2 > 0.94$, $p < 0.001$) using the frequency-pairing method. Annual $Q$ was unchanged at low and high probabilities of non-exceedance (<10 %), but was lower in some years between the 30 % and 60 % probability of non-exceedance (Fig. 4a). Monthly $Q$ was higher than expected at high probability of non-exceedance in February (Fig. 4b); whereas, monthly Q was lower than expected at the high probability of non-exceedance in July (Fig. 4c). Median monthly $Q$ was lower than expected for only Jan (-14.8%) and May (-13.4%) (Table 3). At wetter periods (above median $Q$), monthly $Q$ was lower than expected for several months during and immediately following the growing season (Jun–Oct, Dec; Table 3); whereas, for during Feb–Apr, monthly $Q$ was higher than expected. At drier periods (below median $Q$), February, March and September had lower than expected monthly $Q$ (Table 3). No significant changes in monthly $Q$ distributions were observed in November.

### 3.4  Daily water use (DWU)

Growing season DWU differed among species for any given DBH largely dependent on xylem anatomy (Table 2, Fig. 5). For example, DWU for a for a tree 50 cm DBH could be 6.5 times higher with diffuse-porous xylem compared to ring-porous xylem (Fig. 5). *Liriodendron*





*tulipifera*, *Betula lenta* and *Nyssa sylvatica* had the highest DWU; *Acer rubrum* and *Carya* were
intermediate; and *Quercus alba*, *Q. montana*, and *Q. rubra* had the lowest estimated DWU
compared to all other species for a given diameter (Ford et al., 2011b); *Robinia pseudoacacia*
had higher DWU than *Quercus* or *Carya* (Miniat and Hubbard, unpublished). Models based on
DBH and xylem anatomy explained 55–88 % of the variability in DWU among tree species
(Table 2). For the evergreen understory species, however, DBH explained little variation in
DWU; even though the standard errors were quite low.
Mean growing season DWU for each catchment increased over time, but the treated
watershed showed the greatest increase (Fig. 6a). In the 2010s, the 45 year-old forest in WS6 had
25-43 % higher DWU than the 75+ year-old reference forests (Fig. 6a–d), despite lower leaf area
than the reference watersheds at that time (Fig. 1c). In reference WS14, tree species with diffuse-
porous xylem contributed 48–63 % of the total water use between 1969 (age 35) and 2009 (age
75+), while evergreen shrubs contributed 20–23 %, and tree species with ring-porous xylem
contributed 13 % or less to the total water use (Fig. 6c). Since the grass cover was abandoned in
WS6, tree species with diffuse-porous xylem alone have contributed more than 90 % of the total
daily water use in that watershed (Fig. 6b).

**4 Discussion**

We hypothesized that a shift in species composition and the resulting shift in DWU would
largely explain long-term changes in *Q* in the treated watershed as the forest regenerated
following grass abandonment. We found that forest species composition in the treated watershed
shifted from dominance by species with ring-porous xylem prior to grass conversion to species





with diffuse-porous xylem through old-field succession. With this major shift in species
composition, DWU increased above expected values from 1982 to 2012 in the treated watershed,
and it was much higher than that in the older reference watersheds. These changes in species
composition and DWU correspond with the long-term trend in lower than expected $Q$ over that
time period. Seasonal variation in $Q$ helped to explain this long-term pattern.

**4.1 Vegetation dynamics**

Species composition has changed dramatically in the treated watershed through old-field
succession following the forest-grass-forest transition. Prior to conversion to grass (1958), the
forest was dominated by *Quercus montana* and *Q. coccinea*, similar to the reference watersheds
at that time. After the grass was herbicided, and the forest was allowed to reestablish, the forest
shifted to dominance by *Liriodendron tulipifera* and *Robinia pseudoacacia*. Other studies have
found that shade-intolerant *R. pseudoacacia* and *L. tulipifera* respond and grow rapidly following
clearcutting or other disturbances that create large canopy gaps (Elliott and Swank, 1994; Elliott
et al., 1997, 1998; Shure et al., 2006; Boring et al., 2014). During grass dominance all woody
species were eliminated with spot herbicide application. This treatment killed stump sprouts, and
during forest succession recruitment favored small, wind-dispersed seeds, and discriminated
against large-seeded and slow growing species such as *Quercus*, *Carya*, *Tilia*, and *Aesculus*
(Elliott et al., 1997, 2002). Aboveground biomass approached pretreatment levels after 45 years
of forest growth; however, LAI remained lower than that of the pretreatment or reference
watershed conditions. The lower LAI could be attributed to the differences among species in the
ratio of leaf area per total aboveground biomass and crown structure; where, shade intolerant *R.*



*pseudoacacia* and *L. tulipifera* have lower ratios and concentrate foliage to the uppermost crown
more than intermediate shade-tolerant *Quercus* (Kato et al., 2009).

Many studies have investigated forest growth following harvesting (e.g., Palik et al., 2012;

Boring et al., 2014; Loftis et al., 2014; Stanturf et al., 2014; Boggs et al., 2016), and the
hardwood species composition that succeeds following harvest depends largely on the severity of
disturbance, i.e., partial harvest, retention harvest or clearcutting, as well as the geographical
region (Halpin and Lorimer, 2016). In northern Appalachian forests, *Prunus pensylvanica* and
*Betula papyrifera* are common pioneer species that assume early dominance following
clearcutting (Hornbeck et al., 2014). In central Appalachian forests, *Prunus serotina*, *Acer*
*rubrum*, *Betula lenta*, and *Fagus grandifolia* dominate following extensive harvests
(Kochenderfer, 2006; D′Amato et al., 2015). *Robinia pseudoacacia* and *L. tulipifera*, two species
that recruit and grow rapidly after clearcutting, are much more abundant in the southern
Appalachians (Elliott and Vose, 2011; Boring et al., 2014) than in the central Appalachians
(Kochenderfer, 2006), and are absent in the northern Appalachians (Campbell et al., 2007;
Hornbeck et al., 2014).

**4.2  Species effects on water yield ($Q$) and evapotranspiration (ET)**

We found that annual $Q$ declined and ET increased through old-field succession relative to the
time prior to the grass conversion. After 1980, 13 years following herbicide application, $Q$ was
consistently lower than expected for the next 35 years. $Q$ was reduced by 6.5% averaged over
this time period; however, in 16 of those years, $Q$ was greatly reduced (>50 mm, 9.2 %).  In 2003
and 2015, $Q$ was reduced by 142 mm (16 %) and 113 mm (17 %), respectively. This supports




our hypothesis that changes in ET and $Q$ have occurred as a result of a shift in species
composition. We also found that species effects were seasonal and influenced certain parts of the
flow regime.

The range of changes in $Q$ after treatment suggests that species composition affects storage

and use of water under a wide range of precipitation conditions that play out over monthly and
annual scales. For example, in 2003, when the decrease in $Q$ was greatest (-142 mm; 16%), $P$
was 6% greater than the long-term (1939-2015) average but this followed four years of below
average $P$. Average $P$ for 1999-2002 was 23% below the long-term average. In this case, the
vegetation in old-field succession watershed may have used more of the available water in 2003,
following the dry period, than the vegetation in reference watershed. As a result, less of the
available water served to refill soil storage in the treated watershed compared to the reference
watershed, resulting in a larger predicted decrease in $Q$ in 2003. In 1981 when $Q$ was higher than
expected (+30 mm; 5.5%), $P$ was 29% lower than the long-term average but this followed the
second highest annual $P$ in 1980 (+27% greater than the 1939-2015 average). Much of the excess
rainfall occurred at the end of the 1980 water year and the beginning of water year 1981.
Precipitation during March and April of water year 1980, and May of water year 1981 was
123%, 35%, and 39% greater than the long-term (1939-2015) average for those months,
respectively. Given that the $Q$ for the treated watershed was higher than expected in wetter
months (those above median $Q$) of the dormant season, these wet months resulted in a higher
than expected annual $Q$.

Our monthly analysis showed that changes in ET and $Q$ varied seasonally. First, changes in

monthly distribution of $Q$ suggest that old-field succession and the consequent species changes
have lowered streamflow during the growing season during wetter months. We observed that $Q$




was lower than expected in September during both drier (below median $Q$) and wetter periods
suggesting that changes in soil storage at the end of the growing season highly influences base-
flow. Others have found that forest clearcutting had a longer-lasting influence on streamflow
distribution, even when annual $Q$ returned to baseline conditions within a few years (Burt et al.,
2015; Kelly et al., 2016).

Second, changes in monthly distribution of $Q$ suggest that there is a potential for increased

frequency and severity of high flows in dormant season months under wet conditions. This could
be particularly concerning during severe tropical storms. However, for the Appalachian region
most tropical storms occur later in the year (Sep−Dec) (Holland and Webster, 2007; Smith et al.,
2011). Interestingly, we found lower than expected $Q$ during wetter periods for Sep−Dec months.
If timing of large storms remains unchanged, then shifting species composition from those that
have conservative water use (i.e., ring-porous xylem) to those that are less conservative (i.e.,
diffuse-porous xylem) could mitigate the effects of high flows during large storms.

The observed changes in monthly $Q$ during the dormant season indicate a likelihood of soil

saturation during the wettest periods. Higher than expected $Q$ in the dormant season is likely a
result of lower ET and higher soil moisture at that time of year (Berghuijs et al., 2014; Burt et al.,
2015), rather than reduced infiltration capacity. For example, in an earlier study, Burt and Swank
(1992) reported that the dead grass was not removed following herbicide application on the
treated watershed and so the infiltration capacity remained high throughout 1967 and 1968. More
likely the higher than expected $Q$ in the dormant season is due to the lack of evergreen species in
the treated watershed. Where evergreen species are a component of forested watersheds, they can
affect ET and $Q$ in the dormant season (Brantley et al., 2013, 2015); they transpire during
dormant season months as long as environmental conditions are suitable (Ford and Vose, 2007;



Ford et al., 2011a; Brantley, unpublished data) and they intercept precipitation during the
dormant season because they retain their foliage. Even though evergreens (shrubs + tracheids)
were a relatively small component (13.8 % of total aboveground biomass) of the old-field
succession watershed before treatment, after treatment there were no evergreen shrubs due to the
severity of the treatment. Yet, they remain a component (6.0 % and 15.9 % for WS14 and WS18,
respectively) of the reference watersheds. Thus, evergreen species reduce soil moisture storage
and have the potential to mitigate spring flooding because of their contribution to ET and their
location within riparian zones (Brantley et al., 2015).

Our results demonstrate that species changes largely explain the decreasing trend in $Q$

following old-field abandonment based on modeled growing season DWU over time; and enable
us to assess the effects of forest structure and species composition on $Q$. For example, the
estimates of DWU (Fig. 6) are consistent with the differences in temporal patterns of $Q$ between
the old-field succession WS6 and reference watersheds (Fig. 3). The mean DWU in WS6 was
greater in 1995 than DWU in the reference watersheds in 1969 or 1992, suggesting that $Q$ in
WS6 became less than expected between these years due to altered DWU. Similarly, mean DWU
in the 45 year-old old-field succession WS6 was greater still in 2012 than the 75+ year-old
reference watersheds, WS14 or WS18, in 2010. Indeed, $Q$ was consistently less than expected
during this period, and was significantly less in 32 of the 35 years (including 1995, 2010, and

2012).

Few studies have examined the consequence of shifts in hardwood species composition on

the hydrologic cycle (Swank et al., 2014; Caldwell et al., 2016). Changes in forest composition,
structure and age as well as climate will interact to induce long-term changes in $Q$ from forested
mountain watersheds (von Allmen et al., 2015; Caldwell et al., 2016). We found stronger and



longer lasting decreases in annual and monthly $Q$ through old-field succession, than found by
clearcutting alone followed by forest succession (Reinhart, 1964; Hornbeck, 1973; Swank et al.,
2001, 2014; Troendle et al., 2001; Adams and Kochenderfer, 2014; Hornbeck et al., 2014). For
example, researchers at the Fernow Experimental Forest in West Virginia examined changes in
annual $Q$ following clearcutting (Adams and Kochenderfer, 2014); there, the initial increase in $Q$
returned to pretreatment levels within 3-4 years. In another treated watershed (WS7) in Coweeta
that was allowed to regenerate naturally after the clearcut, there was only one year when
observed $Q$ was significantly lower than predicted (Swank et al., 2014).
The observed changes in monthly and annual $Q$ for the treated WS6 were largely a result of
a rapid response of co-dominant species with less conservative transpiration rates (Wullschleger
et al., 2001; Ford et al., 2012; Boggs et al., 2015; Brantley et al., 2015). Under similar
environmental conditions, both *L. tulipifera* and *R. pseudoacacia* have much higher daily water
use than species with ring-porous and semi ring-porous xylem, such as *Quercus* and *Carya* (Ford
et al., 2011b; Vose et al., 2016a, b). Overall, we estimated that growing season daily water use
increased significantly following old-field abandonment, and it was much higher in the 45 year-
old treated watershed than the 75+ year-old reference watersheds.

**5 Conclusions**

Our long-term results are relevant to land areas that are currently in pasture and those that have
reverted back to forests. In many parts of the world, pastureland and cropland area have
increased since the 1990s as natural landscapes have been converted to agricultural ecosystems
(e.g., Scanlon et al., 2007; Rodriguez et al., 2010); and in other areas agricultural land has been





abandoned (see review Rey Benayas et al., 2007). In general, grass pastures transpire less water
and have lower interception loss than forests resulting in greater $Q$ for this LULC type (e.g.,
Wang et al., 2008; Holdo and Nippert, 2015). In the forest-grass-forest watershed, for two of the
five years when the watershed was in grass cover, $Q$ was equivalent to the pre-conversion
hardwood forest, while for the other three years $Q$ was greater under grass cover. Fertilizer
application in two of the five years resulted in high grass productivity (Hewlett, 1961; Hewlett
and Hibbert, 1966; Burt and Swank, 1992) such that LAI was maximized allowing for ET similar
to that of the reference forested watershed. $Q$ increased initially once herbicide was applied to the
grass, quickly returned to expected levels, and then declined relative to expected levels as the
abandoned old-field was allowed to regenerate to forest.

We found that within a deciduous forest, species identity matters in terms of how much

precipitation leaves the watershed as ET vs. $Q$. Through old-field succession, the treated
watershed was dominated by water demanding species with higher DWU than the pretreatment
forest. We demonstrate that a shift in tree species composition from dominance by species with
ring-porous xylem to dominance by species with diffuse-porous xylem can increase DWU, and
in turn, produce a long-term reduction in $Q$.

Even within unmanaged watersheds, hydrologic parameters are not stationary (*sensu* Milly

et al., 2008; Burt et al., 2015) and subtle changes in species composition can influence $Q$,
particularly in dry years (Caldwell et al., 2016). Species-specific ecohydrological models (e.g.,
Novick et al., 2016) are increasingly vital in predicting long-term changes in ET and $Q$ (Sun, et
al. 2016; Vose et al., 2016a, b). If drought frequency and severity increase as expected (Allen et
al., 2010; Ayres et al., 2014; Peters et al., 2015; Swain and Hayhoe, 2015), then understanding
the interaction of land use, species and climate change on water resources will become even



more important in the future (Grant et al., 2013; Clark et al., 2016; Kelly et al., 2016; Vose et al.,
2016b). As previously outlined as a critical research need (Vose et al., 2016b), our results
provide an example of scaling DWU from tree-level, plots, and small watersheds in order to
understand the species-specific influences on water balance and streamflow dynamics in diverse
Eastern U.S. deciduous forests.

**The Supplement related to this article is available online at doi:-supplement.**

*Data availability.* All data in this manuscript is archived at USDA Forest Service, Southern
Research Station, Coweeta Hydrologic Laboratory, Otto, NC, 28763.

*Acknowledgements.* This study was supported by the USDA Forest Service, Southern Research
Station; and by NSF grants DEB0218001 and DEB0823293 to the Coweeta LTER program at
the University of Georgia. Any opinions, findings, conclusions, or recommendations expressed
in the material are those of the authors and do not necessarily reflect the views of the National
Science Foundation or the University of Georgia. We acknowledge the support of the long-term
climate and hydrologic data network at Coweeta Hydrologic Laboratory as well as many
individuals, past and present, especially Patsy Clinton, Charles Marshall and Stephanie Laseter
for field and climate data collection and processing.

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





Table 1. Characteristics of treated (WS6) and reference (WS14, WS18) watersheds at the Coweeta Hydrologic Laboratory in Otto, NC, USA. Mean annual precipitation ($P$) and water yield ($Q$) based on data collected over water years (WY, May–Apr) from 1934 to 2015.

| Characteristic | Units | Watersheds | | |
| --- | --- | --- | --- | --- |
| | | 6 | 14 | 18 |
| Area | ha | 9.0 | 61.03 | 12.46 |
| Mean elevation | m | 824 | 878 | 823 |
| Mean basin slope | % | 50 | 50 | 55 |
| Aspect | | NW | NW | NW |
| Year of first complete flow record | WY | 1939 | 1938 | 1938 |
| Nearest rain gauge | | SRG41 | SRG41 | SRG96 |
| Mean precipitation ($P$) | mm yr$^{-1}$ | 1843 | 1843 | 2031 |
| Mean water yield ($Q$) | mm yr$^{-1}$ | 866 | 997 | 1021 |
| Mean evapotranspiration ($ET = P - Q$) | mm yr$^{-1}$ | 978 | 845 | 1010 |
| $Q/P$ | | 0.47 | 0.54 | 0.50 |





**Table 2**. Summary of growing season daily water use (DWU, kg day$^{-1}$) models for each xylem

functional group as a function of stem diameter at breast height (DBH, cm); DWU = $b_0$ * DBH$^{b_1}$.

| Xylem Group | N | Min DBH (cm) | Max DBH (cm) | $b_0$ | $b_1$ | Adjusted $R^2$ | SE of estimate |
|---|---|---|---|---|---|---|---|
| Diffuse-porous | 95 | 7.4 | 61.8 | 0.1428 | 1.7676 | 0.70 | 30.3 |
| Evergreen Shrub | 16 | 5.3 | 16.3 | 0.6445 | 0.7002 | 0.00 | 2.5 |
| Ring-porous | 38 | 23.9 | 86.7 | 0.2392 | 1.1488 | 0.55 | 9.1 |
| Semi ring-porous | 18 | 20.2 | 55.7 | 0.0009 | 2.8557 | 0.88 | 8.8 |
| Tracheid | 116 | 9.5 | 67.5 | 0.0005 | 2.8411 | 0.73 | 8.6 |



**Table 3**. Relative changes in monthly water yield ($Q$) for different parts of the cumulative

distribution function for the period May 1979 to Apr 2015 for the treated WS6 using the

frequency-pairing method. Lower and Higher denote direction and significance ($p < 0.05$) of

change, NS = not significant.

| Month | Change in Median Monthly $Q$ (%) | Change in $Q$ below the median $Q$ | Change in $Q$ above the median $Q$ |
|---|---|---|---|
| Jan | Lower (-14.8) | NS | NS |
| Feb | NS | Lower | Higher |
| Mar | NS | Lower | Higher |
| Apr | NS | NS | Higher |
| May | Lower (-13.4) | NS | NS |
| June | NS | NS | Lower |
| Jul | NS | NS | Lower |
| Aug | NS | NS | Lower |
| Sep | NS | Lower | Lower |
| Oct | NS | NS | Lower |
| Nov | NS | NS | NS |
| Dec | NS | NS | Lower |

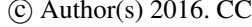


Figures

**Figure 1**. Mean (±SE bars) **(a)** aboveground biomass, **(b)** foliage biomass, and **(c)** leaf area index (LAI) for the treated WS6 and reference (WS14, WS18) watersheds over time.

**Figure 2**. Percent (±SE bars) aboveground biomass for the xylem functional groups (diffuse-porous, ring-porous, semi-ring porous, tracheid, and evergreen shrub) in the **(a)** treated WS6, **(b)** reference WS14, and **(c)** reference WS18 over time.

**Figure 3**. Change in water yield ($Q$, $D = Q_T - (\hat{Q}_T; M1, M2 = 0)$) for the treated WS6 over time (bars). Solid lines are the standard errors of the mean prediction. We used the paired-watershed approach with WS18 as the reference. The year of harvest, conversion to Kentucky-31 fescue grass (*Festuca octiflora*) cover, fertilize, herbicide, and abandonment to allow forest regeneration are denoted by dashed lines.

**Figure 4.** Changes in the cumulative distribution function (CDF) expressed as historic probability of non-exceedance for **(a)** annual water yield ($Q$), and monthly $Q$ for **(b)** February and **(c)** July. * ($p < 0.05$) and ** ($p < 0.01$) denote years in the distribution functions when $Q$ was significantly lower or higher than predicted.

**Figure 5**. Growing season daily water use of tree species by xylem functional group (diffuse-porous, ring-porous, semi-ring porous, evergreen shrub and tracheid) and DBH (diameter at 1.37 m above ground).

**Figure 6**. **(a)** Mean (±SE bars) growing season daily water use (DWU) versus forest age in the treated WS6 and reference (WS14, WS18) watersheds; **(b)** DWU versus forest age in treated WS6 by xylem functional group (diffuse-porous, ring-porous, semi-ring porous, evergreen shrub, and tracheid); **(c)** DWU versus forest age in reference WS14 by xylem functional group; and **(d)** DWU versus forest age in reference WS18 by xylem functional group.





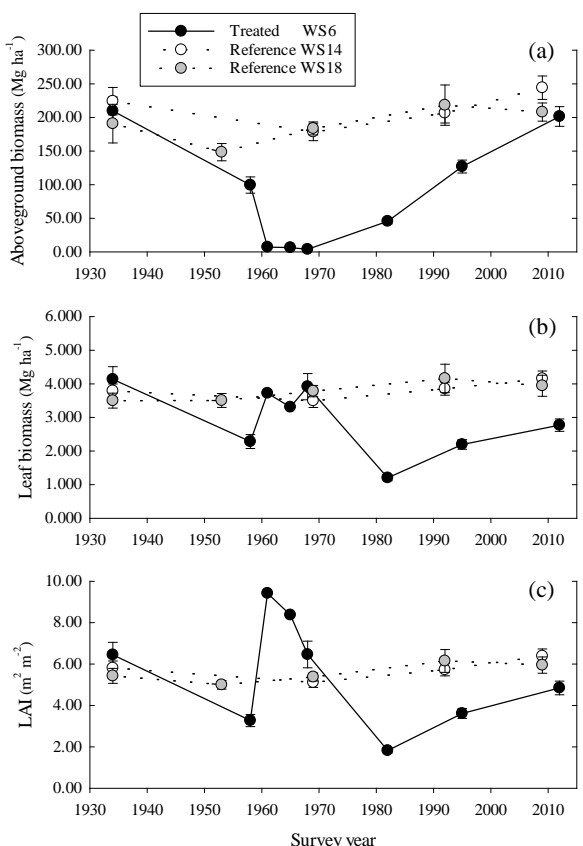

Figure 1



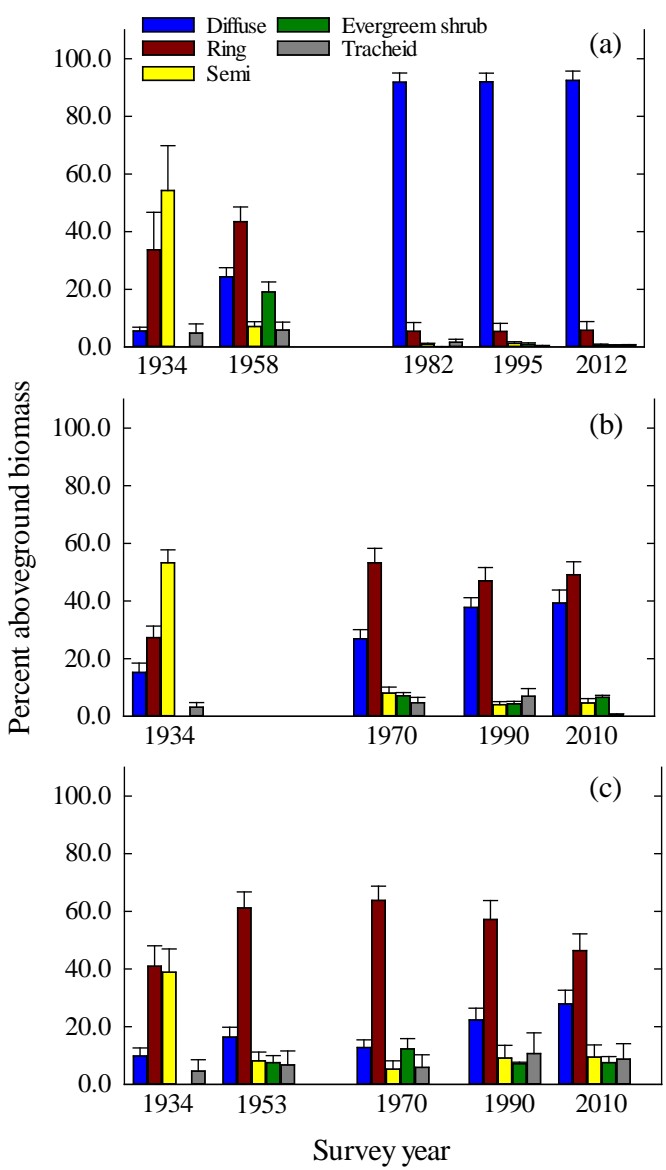

Figure 2

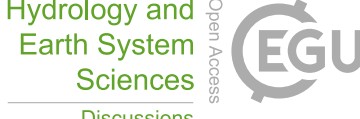

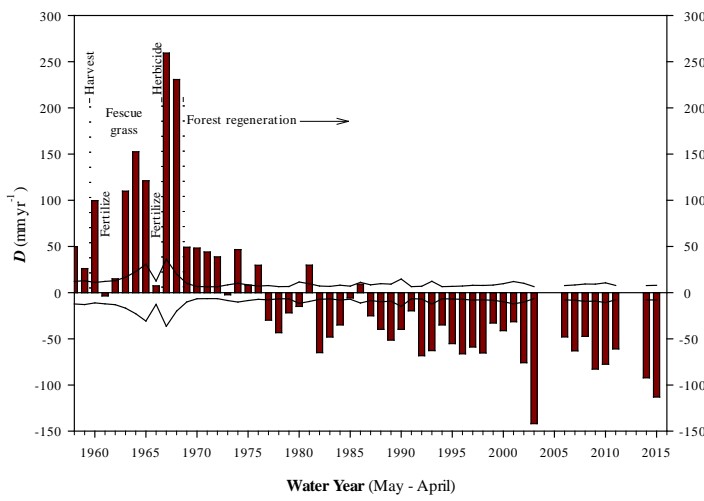

Figure 3





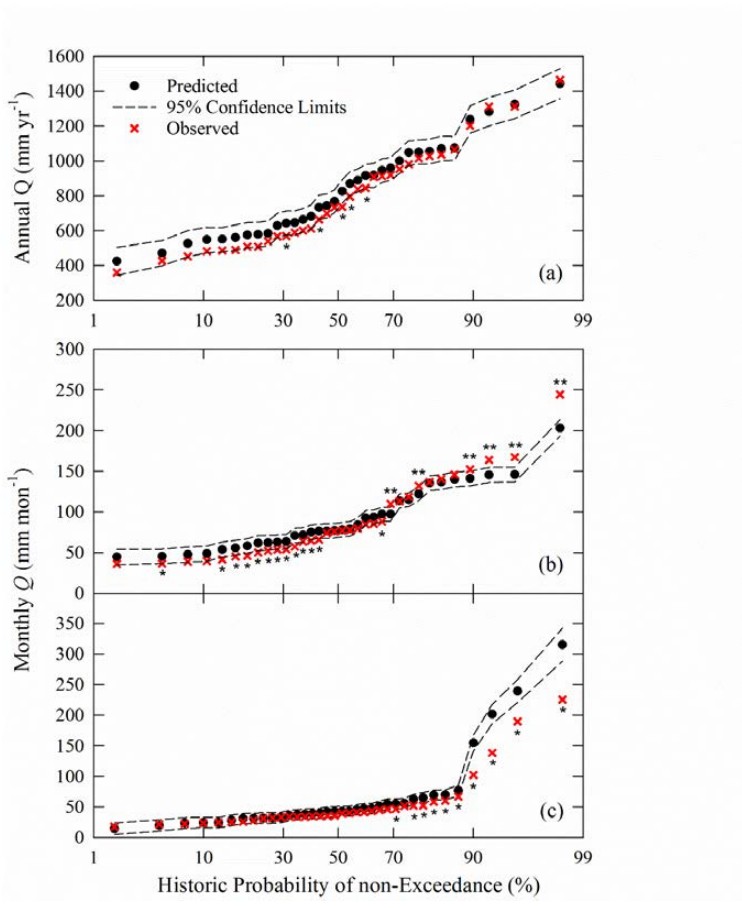

Figure 4





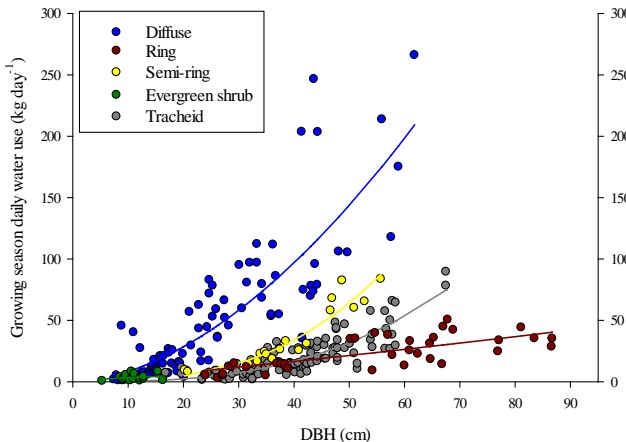

Figure 5





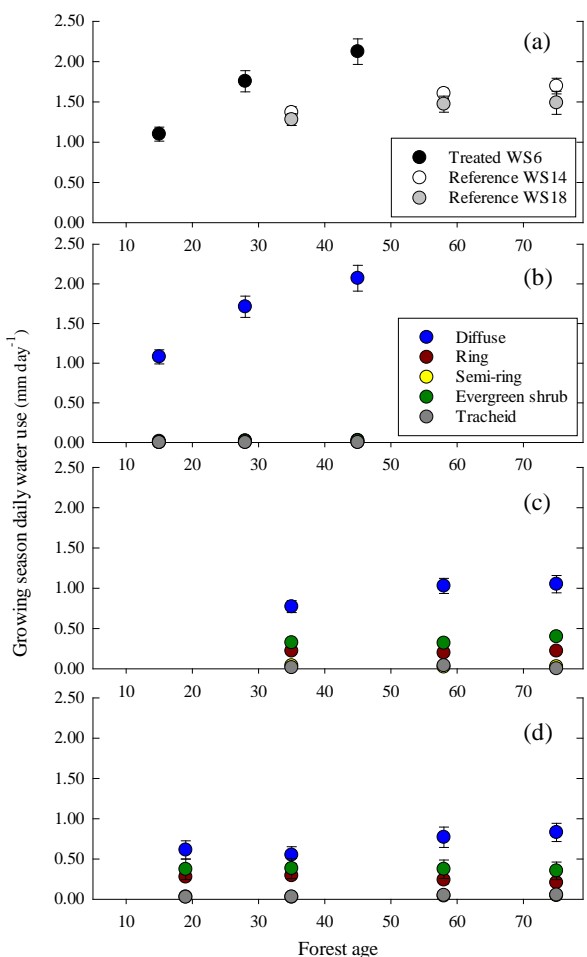

Figure 6