# Peer review of "Water yield following forest−grass−forest transitions"

_Hydrology and Earth System Sciences, 2016_

## Referee Comment (RC1) · Anonymous Referee #1 · 23 Nov 2016

Comments to Author

Summary: This paper investigates the potential effects of land use and land cover changes on water yield (Q) and evapotranspiration (ET) by focusing on shifts in tree species composition during old-field succession. From a long data set (about 80 years) the authors observed a management induced change in vegetation from forest dominated by Quercus and Carya to grass and finally to regrown forest dominated by Liriodendron and Acer. These shifts were evident in the Q data. The conversion of forest to grass resulted in increases in Q, similar to previous studies that have studied the effects of clear-cutting on Q. The regrowth of forest, however, resulted in a decrease in Q and the shift in tree species composition resulted in Q becoming lower than in the original forest. The authors claim that this shift in Q was a result of changes in ET because of differences in water use among tree species. Liriodendron and Acer

have a higher water use than Quercus and Carya. The authors also observed monthly changes in Q, especially in wetter months.

Contributions: Knowledge about how vegetation influences ET and Q is still not well understood. This research is therefore timely and important, and a useful step towards a better understanding of these ecohydrological issues. The authors present a good data set that in itself is worthy of publication; I can imagine that many scientists in the field could make use of these data. The text is relatively well written and logically organized text. Still, I do have some remarks detailed in a number of general and technical comments below.

General comments:

One thing I found curious was that the focus is entirely on Q since ET is hardly mentioned in the ms. The authors claim to have calculated ET but the only data I found was a long term average calculated from long term precipitation and runoff. I have no doubt the authors actually calculated ET (and there is very brief description of how this was done in the Methods) but I think these results should be presented (could be in the supplementary info) for the reader to be able to compare changes in Q and ET.

I found the description of some of the methods to be unclear and too brief. This is unfortunate, since the ms otherwise is well-organized and well written. The lack of a thorough methods description makes it difficult to value the validity of the results. I kept thinking "this is nice, but since I cannot fully appreciate the methodology I cannot judge if the conclusions are correct and logically follow from the results". Some examples may illustrate my point:

-Section 2.3.1: how was basal area, aboveground biomass and LAI estimated? Equations here could clarify this.

-Section 2.3.2: there is no explanation of the model (or are there several models? – it was not clear to me) used to estimate Q without treatment effects (equation 1). Was

this model developed in this project (and if so, based on what? Derivation, please) or is this something the authors have used previously (reference, please)?

-Section 2.3.2: the difference between QT and QT hat was not clear to me at first. I think I got it after reading the section several times, but then I am probably doing some guessing

-Section 2.3.3: some of the data before the first treatment was used to find regression models between Q in the treatment watershed and the reference watershed, but why was not all data from the pre-treatment period used? Seems arbitrary.

-What is the rationale of using both equation 1 and equation 4? Do they model similar/same data?

These methods are key to understanding the paper and need to be thoroughly described and explained. Schematic figures could help (e.g. to describe the concepts of chronological pairing and frequency pairing), if applicable.

The authors report monthly deviations in Q, but I miss a thorough discussion about potential causes of some aspects of these results. There is a discussion about Q being lower than expected during wet months during the growing season, and I agree with the authors here, but why is there no effect during other months in the growing season (i.e. months that are not classified as wet). Should you not expect to see the same pattern in those months? And I found no convincing explanation to why Q is higher than expected during wet months during the dormant season. Also, according to figure 4b, Q is lower than expected at around average wetness during the dormant season. Why is that?

Other small comments:

-A map of the watersheds, rain gauges, weirs/flumes etc. could be useful

-Finally a very small comment, and I admit being perhaps a little too strenuous, but I found the usage of semicolon a bit strange from time to time. As far as I recall, semicolon is used to connect two independent clauses, or when listing units that include commas.

Altogether, this manuscript is a valuable addition to the scientific field and I support its publication in HESS. The science is as far as I can tell sound but the science communication could be improved. I recommend major revisions of the manuscript before the editor considers publication of the manuscript.

Technical comments:

Line 35: here you say that ET increase by 3.4% but in the main text it is 4.5%; which is it?

Line 62: remove "the" before Brantley

Line 63: why do you use a plus sign after 20%? I suggest you use > if you mean more than.

Line 67: This is not a recommended usage of semicolon. Semicolon is used to connect two independent clauses, or to separate units in lists where each unit contains one or more commas. It should very rarely be combined with conjunctions such as "and".

Line 86: is "by" the correct preposition here?

Line 93: I found the word "studied" in this sentence strange.

Line 100: and I guess eastern in this case refers to eastern USA? Could be worth clarifying that

Line 101: Another use of semicolon that seems strange. While, in the meaning whereas, i.e. a comparison, should be separated by a comma rather than a semicolon

Line 102: I found this sentence unclear and I think that is partly because you have "rates" here. Are rates lower than L. tulipifera? Should the word "rates" be placed before the other species (Acer and Betula)?

Line 127-129: Why this hypothesis about monthly effects? There is nothing in the Introduction that prepares the reader for this hypothesis. You need some background or theory to argue for why this would be a relevant hypothesis. Now it feels like you added this hypothesis based on the results (which is odd!).

Line 138-140: A map of the relative positions and sizes of the watersheds, positions of rain gauges etc. would be nice.

Line 152: What do you mean by last decade here? The 1940s or 2000s or something else?

Line 168: Repetition of grass species – I do not think this is necessary.

Line 195: Remove "at diameter"

Line 201-203: How were basal area, aboveground biomass and LAI calculated? Equations could be useful (could even go into the supplementary info)

Line 225: model or models?

Line 226: What is the rationale of this model? Where does it come from? Derivation? Reference? You need to explain/derive this model. As it is now, I am left in confusion.

Line 239: P is not used in the equation. Is it necessary?

Line 244: Should this be Q(QT) or is there a Q too much? What is QT? Is QT observed Q in the treated watershed and QT hat the estimated Q in the treated watershed without treatment effects (estimated from the reference watershed)?

Line 251-252: I think I understand what you mean but I found this sentence unclear. Also, "basin" should be spelled with lower-case letters.

Line 264: Why did you use these years and not all pre-treatment years? Seems arbitrary.

Line 267: What does equation 4 do, that equation 1 cannot?

Line 279: I understand m and n, but where do the constants come from (0.40 and 0.20)?

Line 284: there is one parenthesis too much in the second term inside the square root sign. Should it be Ym in both terms? Then how do Var1 and Var2 differ?

Line 286: I think there is an equation number missing here

Line 287: "analyses" instead of "analysis"

Line 296: "...numerous species and stem diameter sizes..."

Line 298: "fitted" instead of "fit"?

Line 366: It would be nice to see these results. Also, the number here differs from the number in the Abstract

Line 375: Why >0.94 and not =0.94?

Line 379: no semicolon

Line 382: no semicolon

Line 383: remove "during" (or for)

Line 390: repetition of "for a" – remove one occurrence

Line 394: I suggest you replace the semicolon with a full stop.

Line 398: replace semicolon with comma

Line 411-413: Would it be possible to correlate the annual species weighted DWU with annual Q over time?

Line 416: What do you mean by "expected values" here? You have not calculated any expected DWU values.

Line 437: replace semicolon with comma

Line 464: Can you really draw this conclusion? You have no data on water storage, so you basically assume Q = f(S) and thus that S decreases when Q decreases. Would not this assumption contradict your assumptions when calculating ET? You assume that the change in storage is 0 over time when calculating ET, but in your further discussion (lines 466-480) you discuss carryover effects due to e.g. drought (i.e. less P than usual). If you calculate ET as P-Q you assume no change in storage during that time period.

Line 469: insert a "the" before "old-field succession watershed"

Line 470: "reference watersheds" or "the reference watershed"

Line 478-480: and why is Q higher in the treated watershed in wetter months during the dormant season? I found no explanation or speculation.

Line 481-483: Why is this effect only evident during wet months and not during e.g. months with normal wetness?

Line 516: no semicolon

Line 522: do you mean >75 year-old?

Line 538-545: I found most of this section repetitious. Is this section really necessary?

Line 552: no semicolon

Table 1: It is interesting that P differs between the treatment and reference watersheds (and the difference is about 10%) – are the rain gauges within the watersheds? If not, how far away are the rain gauges?

Table 2: Was the R2 for the evergreen really 0? If so, was that model ever used?

Figure 1: In the methods you mention estimating aboveground biomass but not leaf biomass – how was this estimated?

---

## Referee Comment (RC2) · Anonymous Referee #2 · 27 Dec 2016

Overall this is a good paper from a distinguished team using long-term data sets with a history of quality measurement. That's the good part. The bad part is that the paper does take a lot of following and reading, and after going through it a number of times, I'm still not entirely sure about the methodology. I think that this may reflect on the reviewer more than the writers, but the paper is a bit uncompromising in its terse presentation of information; to my mind that detracts a little from what is . overall, a fine piece of work. Some of my comments relate to the need to perhaps "help" the readers a little.

I think that the paper would be improved a little by better graphics. Firstly a picture or two of WS6 at various stages would help. Similarly, a "time-line" of its treatments would also be useful. A small map showing the various watersheds would be good, too.

I presume that the authors are trying to suppress a certain amount of detail – such as the development of Equation 1 (which presumably goes back a long way). For the non-hydrologists such an equation would be pretty enigmatic; I guess it is a judgement call for the authors, but it is asking readers a lot to swallow this at one gulp, so to speak. Ditto the frequency-pairing method.

The authors raise the very interesting point about non-stationary "controls" during these long-term paired watershed studies. It is probably the weakest point of this approach once they get past four or five decades (but when these were put in, who envisaged them going that long?). The difficulty is that I am not sure what one might do on this matter. Perhaps the authors could talk about this a bit more?

So overall, it is a fine paper but a bit hard-going in its methods. The "discussion" probably needs a bit of tightening since it is to some extent speculative. I think that it would be worthy of a longer paper in which the methods are teased out and there is more explanatory hydrograph detail. As an aside, I have always found it annoying that Dunford and Fletcher (1941) commented on the loss of the diurnal variation (if I recall correctly) but that no one ever seems to have looked at this again (i.e. how long did it take to reappear), and wonder if such a longer paper might also include indicators like this.

---

## Author Comment (AC1) · 5 Jan 2017

Elliott et al. Anonymous Referee #1

Comments to Author

Summary: This paper investigates the potential effects of land use and land cover changes on water yield (Q) and evapotranspiration (ET) by focusing on shifts in tree species composition during old-field succession. From a long data set (about 80 years) the authors observed a management induced change in vegetation from forest dominated by Quercus and Carya to grass and finally to regrown forest dominated by Liriodendron and Acer. These shifts were evident in the Q data. The conversion of forest

to grass resulted in increases in Q, similar to previous studies that have studied the effects of clear-cutting on Q. The regrowth of forest, however, resulted in a decrease in Q and the shift in tree species composition resulted in Q becoming lower than in the original forest. The authors claim that this shift in Q was a result of changes in ET because of differences in water use among tree species. Liriodendron and Acer have a higher water use than Quercus and Carya. The authors also observed monthly changes in Q, especially in wetter months.

Contributions: Knowledge about how vegetation influences ET and Q is still not well understood. This research is therefore timely and important, and a useful step towards a better understanding of these ecohydrological issues. The authors present a good data set that in itself is worthy of publication; I can imagine that many scientists in the field could make use of these data. The text is relatively well written and logically organized text. Still, I do have some remarks detailed in a number of general and technical comments below.

General comments: One thing I found curious was that the focus is entirely on Q since ET is hardly mentioned in the ms. The authors claim to have calculated ET but the only data I found was a long term average calculated from long term precipitation and runoff. I have no doubt the authors actually calculated ET (and there is very brief description of how this was done in the Methods) but I think these results should be presented (could be in the supplementary info) for the reader to be able to compare changes in Q and ET.

Response: We included a figure similar to Fig. 3, but showing the difference in ET instead of Q (Fig. S2 in Supporting Information). It is basically opposite in sign from the change in Q. And see response below for lines 222-261.

I found the description of some of the methods to be unclear and too brief. This is unfortunate, since the ms otherwise is well-organized and well written. The lack of a thorough methods description makes it difficult to value the validity of the results. I kept

[Figure]

thinking "this is nice, but since I cannot fully appreciate the methodology I cannot judge if the conclusions are correct and logically follow from the results". Some examples may illustrate my point:

Response: We have provided more detail in the methods and responded to each of the illustrated points below.

-Section 2.3.1: how was basal area, aboveground biomass and LAI estimated? Equations here could clarify this.

Response: We included the references for published allometric equations that were developed on-site at Coweeta Hydrologic Laboratory (lines 201-204).

-Section 2.3.2: there is no explanation of the model (or are there several models? – it was not clear to me) used to estimate Q without treatment effects (equation 1). Was this model developed in this project (and if so, based on what? Derivation, please) or is this something the authors have used previously (reference, please)?

Response: to clarify the model and provide previous use (references) we revised text on lines 222-261 as follows:

"We modeled WS6 annual Q and ET as a function of WS18, incorporating the effect of grass conversion and reforestation treatments over time. Annual Q was computed on a May–April water year to minimize the effects of year-to-year changes in storage, as soils are generally at their wettest by the beginning of May. The empirical chronological-pairing model was fit using PROC NLIN (SAS v9.4, SAS Institute, Cary, NC) and had the following form: Q ÌĆ_T=a+bQ_R+eM1t1+[M2c(h-1/(1+ãĂŰexpãĂŮˆ(-t2) ))] (1) where, Q ÌĆ_T = predicted Q from treated watershed WS6 (mm yr-1), QR = measured Q from reference watershed WS18 (mm yr-1), M1 = management representing grass conversion; M1 = 1 for water years between and including 1960 and 1966, M1 = 0 otherwise, t1 = time since grass fertilization; t1 = water year – fertilization year for water years between and including 1960 and 1966 where fertilization years

[Figure]

include water years 1959, 1961, and 1966, t1 = 0 otherwise, M2 = management representing reforestation after grass conversion; M2 = 1 for water years greater than or equal to 1967, M2 = 0 otherwise, t2 = time since reforestation after grass conversion; t2 = water year – 1967 for water years greater than or equal to 1967, t2 = 0 otherwise, P = annual precipitation (mm yr-1) a, b, c, e, h are fitted parameters. This overall modeling approach has been used in prior studies to assess the impact of forest management on Q (Ford et al., 2011; Kelly et al, 2016). The a+bQ_R term in EQ1 reflects the relationship between reference and treatment watersheds assuming no treatment. The increasing linear Q response (eM1t1 term in EQ1) accounts for the decline in annual grass production and water use after fertilization as noted by Hibbert (1969). The M2c(h-1/(1+ãĂŰexpãĂŮ^(-t2) )) term in EQ1 accounts for the exponential decline in Q as the forest regenerates that has been observed in numerous paired watershed experiments (Swank et al., 1988). As in Ford et al. (2011), we define the Q treatment response, DQ, as the difference between the observed Q in the treated watershed (QT) and that predicted by the model assuming no treatments had taken place (Q ÌĆ_T) : D_Q=Q_T-(Q ÌĆ_T; M1,M2=0). (2) The proportion of the variability explained by the model was quantified using the ratios of the error-to-total sum of squares and the total-to-error degrees of freedom as: R_adjusted^2=1-ãĂŰSSãĂŮ_E/ãĂŰSSãĂŮ_T × ãĂŰdfãĂŮ_T/ãĂŰdfãĂŮ_E . (3) Parameter estimates were interpreted as statistically significant at ïĄą = 0.05. Observed annual ET was computed as precipitation (P) – QT while expected ET with no treatment was computed as P - Q ÌĆ_T, both assuming the largely impermeable bedrock underlying the Basin that results in negligible deep groundwater losses (Douglass and Swank, 1972). Watershed P was estimated using a nearby eight inch (20.3 cm) National Weather Service standard rain gauge, SRG 96 (Laseter et al., 2012). The ET treatment response, DET, is then: D_ET=[ãĂŰP-QãĂŮ_T ]-([ãĂŰP-Q ÌĆãĂŮ_T ]; M1,M2=0) (4)."

-Section 2.3.2: the difference between QT and QT hat was not clear to me at first. I think I got it after reading the section several times, but then I am probably doing some guessing

Response: The above revisions should clarify the use of "QT and QT hat".

-Section 2.3.3: some of the data before the first treatment was used to find regression models between Q in the treatment watershed and the reference watershed, but why was not all data from the pre-treatment period used? Seems arbitrary. -What is the rationale of using both equation 1 and equation 4? Do they model similar/same data? These methods are key to understanding the paper and need to be thoroughly described and explained. Schematic figures could help (e.g. to describe the concepts of chronological pairing and frequency pairing), if applicable.

Response: Equation 1 and EQ 4 (now Equation 5) do use the same data. We provide the rationale on lines 209-217. There were small gaps in the flow record during the pretreatment period. The years listed, on line 272, are those where we had data.

The authors report monthly deviations in Q, but I miss a thorough discussion about potential causes of some aspects of these results. There is a discussion about Q being lower than expected during wet months during the growing season, and I agree with the authors here, but why is there no effect during other months in the growing season (i.e. months that are not classified as wet). Should you not expect to see the same pattern in those months? And I found no convincing explanation to why Q is higher than expected during wet months during the dormant season. Also, according to figure 4b, Q is lower than expected at around average wetness during the dormant season. Why is that?

Response: We discuss why Q is higher than expected during wet months during the dormant season on lines 505-522, here we suggest that because WS6 after treatment does not have rhododendron or hemlock (evergreen species) ET would be lower than in the reference watershed where they remain a component. For example, "More likely the higher than expected Q in the dormant season is due to the lack of evergreen species in the treated watershed." On lines 510-512.

Other small comments: -A map of the watersheds, rain gauges, weirs/flumes etc. could

be useful Response: We included a map of the Coweeta Basin with landmarks noted. Now Figure 1, referenced on line 141.

-Finally a very small comment, and I admit being perhaps a little too strenuous, but I found the usage of semicolon a bit strange from time to time. As far as I recall, semicolon is used to connect two independent clauses, or when listing units that include commas.

Response: We have corrected semicolon use, and minor comments below as follows.

Altogether, this manuscript is a valuable addition to the scientific field and I support its publication in HESS. The science is as far as I can tell sound but the science communication could be improved. I recommend major revisions of the manuscript before the editor considers publication of the manuscript.

Response: We have improved the science communication by responded to each of the comments provided by the reviewer.

Technical comments: Line 35: here you say that ET increase by 3.4% but in the main text it is 4.5%; which is it?

Response: Changed to 4.5% on line 35

Line 62: remove "the" before Brantley

Response: removed 'the' on line 62.

Line 63: why do you use a plus sign after 20%? I suggest you use > if you mean more than.

Response: removed the plus sign, now reads as > 20%, line 63.

Line 67: This is not a recommended usage of semicolon. Semicolon is used to connect two independent clauses, or to separate units in lists where each unit contains one or more commas. It should very rarely be combined with conjunctions such as "and".

Response: replaced semicolon with a comma on line 68.

Line 86: is "by" the correct preposition here?

Response: replaced "by" with "of" on line 87.

Line 93: I found the word "studied" in this sentence strange.

Response: removed the word "studied" on line 94.

Line 100: and I guess eastern in this case refers to eastern USA? Could be worth clarifying that

Response: replaced "temperate deciduous" with "Eastern U.S. deciduous" on line 97.

Line 101: Another use of semicolon that seems strange. While, in the meaning whereas, i.e. a comparison, should be separated by a comma rather than a semicolon

Response: placed a comma after "while," on line 102.

Line 102: I found this sentence unclear and I think that is partly because you have "rates" here. Are rates lower than L. tulipifera? Should the word "rates" be placed before the other species (Acer and Betula)?

Response: changed to "while, rates of Acer rubrum L. and Betula lenta L., also common diffuse-porous species, are lower than L. tulipifera." on lines 102-103.

Line 127-129: Why this hypothesis about monthly effects? There is nothing in the Introduction that prepares the reader for this hypothesis. You need some background or theory to argue for why this would be a relevant hypothesis. Now it feels like you added this hypothesis based on the results (which is odd!).

Response: added to lines 65-66 "Brantley et al. (2013) also suggested that a change in forest composition with less evergreen hemlock relative to deciduous trees could result in an increase in Q in winter months."

Line 138-140: A map of the relative positions and sizes of the watersheds, positions of

rain gauges etc. would be nice.

Response: We added a figure with the map of the Coweeta Basin and watersheds, now Figure 1, see response above.

Line 152: What do you mean by last decade here? The 1940s or 2000s or something else? Response: to clarify, we changed to "in the 2000s" on line 152.

Line 168: Repetition of grass species – I do not think this is necessary. Response: removed "Festuca octiflora" from this sentence, on line 167.

Line 195: Remove "at diameter" Response: removed "at diameter" on line 194.

Line 201-203: How were basal area, aboveground biomass and LAI calculated? Equations could be useful (could even go into the supplementary info)

Response: We chose to not include these published allometric equations as they are easily obtainable, we included references. Basal area is a simple geometric function, the area of a circle ($\pi r2$) assuming the stem is circular, it is used commonly in ecology and forestry.

Line 225: model or models?

Response: "model"

Line 226: What is the rationale of this model? Where does it come from? Derivation? Reference? You need to explain/derive this model. As it is now, I am left in confusion.

Response: We included references and more detail about the model. See response under Section 2.3.2 above

Line 239: P is not used in the equation. Is it necessary?

Response: P was not used in the equation, we deleted line 239.

Line 244: Should this be Q(QT) or is there a Q too much? What is QT? Is QT observed Q in the treated watershed and QT hat the estimated Q in the treated watershed without

treatment effects (estimated from the reference watershed)?

Response: this was rewritten, see response to general comment above under Section 2.3.2.

Line 251-252: I think I understand what you mean but I found this sentence unclear. Also, "basin" should be spelled with lower-case letters.

Response: Sentence was clarified above, we changed to lower case for "basin" on line 257.

Line 264: Why did you use these years and not all pre-treatment years? Seems arbitrary.

Response: There were gaps in the flow record during the pretreatment period. The years listed are those where we had data.

Line 267: What does equation 4 do, that equation 1 cannot?

Response: This is the difference between chronological pairing (EQ1) and frequency pairing (EQ4). The rationale for using both equations was described on lines 209-221: "The chronological pairing approach allowed us to create a time series of estimated change in annual Q and ET over the period of record and to relate these changes to both the treatment and to climate. In addition, this analysis allowed us to determine when a consistent change in Q began, enabling us to establish the time period of interest for the frequency pairing. The frequency pairing approach allowed us to compare the post-treatment distribution of monthly and annual Q to that of the pretreatment period."

Line 279: I understand m and n, but where do the constants come from (0.40 and 0.20)?

Response: the constants are part of the equation, where the right-hand side of (EQ6) is the approximately quantile-unbiased Cunnane plotting position [Stedinger et al., 1993].

We cite Cunnane, 1978 and Stedinger et al., 1993 for this equation on lines 288-289, "This function provided an empirical estimate of the quantile for a given flow value (Cunnane, 1978; Stedinger et al., 1993)." Brantley et al., 2015 used this analysis in a recent paper, it was also used in Alila et al. 2009. We included citations on lines 299-300.

Line 284: there is one parenthesis too much in the second term inside the square root sign. Should it be Ym in both terms? Then how do Var1 and Var2 differ?

Response: We removed the extra parenthesis. Yes, it should be [Ym] in each term, it is VAR1 and VAR2 that differ, these are explained on lines 292-294, "We used a pair of Monte-Carlo simulations to estimate the variability associated with the predictive uncertainty in equation (Var1), and the uncertainty associated with the sampling variability at each rank (Var2)." References are also cited for this analysis (Alila et al., 2009; Green and Alila, 2012; Brantley et al., 2015) on lines 299-300.

Line 286: I think there is an equation number missing here Line 287: "analyses" instead of "analysis"

Response: We now have all equations numbered in sequence.

Line 296: ". . . numerous species and stem diameter sizes. . ." Response: changed to " . . . numerous species and stem diameter sizes.." on lines 304-305.

Line 298: "fitted" instead of "fit"? Response: we changed to "fitted" on line 307.

Line 366: It would be nice to see these results. Also, the number here differs from the number in the Abstract

Response: We have included a figure of ET (Fig. S2), we have changed the value in the abstract to the correct value of 4.5 %.

Line 375: Why >0.94 and not =0.94? Line 379: no semicolon Response: corrected to "r2 = 0.94" on line 383.

Line 382: no semicolon Response: corrected

Line 383: remove "during" (or for) Response: removed "during" on lines 389 and 390.

Line 390: repetition of "for a" – remove one occurrence Response: removed "for a given diameter" on line 402.

Line 394: I suggest you replace the semicolon with a full stop. Line 398: replace semicolon with comma Response: we replaced the semicolon with a period on line 402, and we replaced the semicolon with a comma on line 405.

Line 411-413: Would it be possible to correlate the annual species weighted DWU with annual Q over time?

Response: This is a good suggestion, we considered correlating DWU with ET as part of the manuscript. However there are few data points of DWU (e.g., four for WS6), thus we decided that a correlation on so few data would not very be meaningful.

Line 416: What do you mean by "expected values" here? You have not calculated any expected DWU values. Response: we removed "expected values' on line 424.

Line 437: replace semicolon with comma Response: replaced

Line 464: Can you really draw this conclusion? You have no data on water storage, so you basically assume Q = f(S) and thus that S decreases when Q decreases. Would not this assumption contradict your assumptions when calculating ET? You assume that the change in storage is 0 over time when calculating ET, but in your further discussion (lines 466-480) you discuss carryover effects due to e.g. drought (i.e. less P than usual). If you calculate ET as P-Q you assume no change in storage during that time period.

Response: We can infer that the changes in forest age and species composition have had an impact on storage recharge dynamics through differences in water use. On lines 472-480, we speculate that the change in the vegetation in the treatment watershed

affected the rate at which storage was recharged when compared to the reference watershed over short time periods (e.g. year to year). The calculation of ET as P-Q assumes negligible change in storage over longer time periods. This is a common assumption that has been used for decades in forest hydrological research. The longer the time period, the more negligible the change in storage. Where we report changes in ET (lines 371-375), it was calculated over the 35 years from 1980-2015 thus change in storage could be assumed to be negligible. Over shorter time periods (e.g., one year to another), changes in storage could occur if one year was particularly wetter or drier than the other, and differences in water use by vegetation can affect storage dynamics.

Line 469: insert a "the" before "old-field succession watershed" Line 470: "reference watersheds" or "the reference watershed"

Response: added "the" on line 477, and "reference watersheds" on line 478, since we are talking about vegetation in this case.

Line 478-480: and why is Q higher in the treated watershed in wetter months during the dormant season? I found no explanation or speculation. Response: See response above under general comments.

Line 481-483: Why is this effect only evident during wet months and not during e.g. months with normal wetness? Response: See response above under general comments.

Line 516: no semicolon Response: removed semicolon on line 523.

Line 522: do you mean >75 year-old? Response: changed to > 75 year-old.

Line 538-545: I found most of this section repetitious. Is this section really necessary? Line 552: no semicolon

Response: now lines 534-545, compares results from this study to other studies where clearcutting was used but the forest was allowed to regenerate immediately and naturally, it seems relevant to compare the magnitude of change in Q in this study compared

to others.

Table 1: It is interesting that P differs between the treatment and reference watersheds (and the difference is about 10%) – are the rain gauges within the watersheds? If not, how far away are the rain gauges?

Response: The rain gauges are shown in the new map figure (Fig. 1). SRG41 paired with WS6 and WS14 is on the boundary between these two watersheds, SRG96 paired with WS18 is approximately 275m from the WS18 western boundary. Precipitation in the Coweeta basin increases from east to west due to elevation and interactions between predominant weather patterns and orographic effects (Swift, 1988). As a result, precipitation is higher in WS18 than in WS6 and WS14. The paired watershed approach accounts for differences in precipitation between reference and treatment watersheds.

Table 2: Was the R2 for the evergreen really 0? If so, was that model ever used?

Response: We did not use the model for evergreen, rather we used the mean DWU value, on lines 318-321, "For the two understory evergreen species, Kalmia latifolia and Rhododendron maximum, we applied the mean DWU value from 16 instrumented shrubs because DWU models based on DBH alone provided limited predictive power (Table 2)."

Figure 1: In the methods you mention estimating aboveground biomass but not leaf biomass – how was this estimated?

Response: now Figure 2, we also used allometric equation for leaf biomass, we added 'leaf biomass' to the sentences on lines 200-204, "Median DBH values were used to calculate basal area, aboveground biomass, leaf biomass, and LAI. We used species-specific allometric equations developed on-site to estimate the aboveground biomass, leaf biomass, and LAI contribution of each species in each watershed (McGinty, 1972; Santee and Monk, 1981; Martin et al., 1998; Ford and Vose, 2007; B.D. Kloeppel,

unpublished data; C.F. Miniat, unpublished data)."
Overall this is a good paper from a distinguished team using long-term data sets with a history of quality measurement. That's the good part. The bad part is that the paper does take a lot of following and reading, and after going through it a number of times, I'm still not entirely sure about the methodology. I think that this may reflect on the reviewer more than the writers, but the paper is a bit uncompromising in its terse presentation of information; to my mind that detracts a little from what is overall, a fine piece of work. Some of my comments relate to the need to perhaps "help" the readers a little. I think that the paper would be improved a little by better graphics. Firstly a picture or two of WS6 at various stages would help. Similarly, a "time-line" of its treatments would also be useful. A small map showing the various watersheds would be good, too.

Response: We have included a Figure with the map of the watershed locations, weirs, and rain gages, now Figure 1. A time frame of WS6 treatments is provided in Table S1, supporting information. On line 153 "The disturbance regime in WS6, the treated watershed, was extensive (Table S1)."

I presume that the authors are trying to suppress a certain amount of detail – such as the development of Equation 1 (which presumably goes back a long way). For the non-hydrologists such an equation would be pretty enigmatic; I guess it is a judgement call for the authors, but it is asking readers a lot to swallow this at one gulp, so to speak. Ditto the frequency-pairing method.

Response: We have retained all of the Equations in the paper because they are important in understanding the hydrology methodology. The methods are also well referenced. We revised the section on methods based on comments from Reviewer #1 to provide addition detail: to clarify the model and provide previous use (references) we revised text on lines 222-261 as follows:

"We modeled WS6 annual Q and ET as a function of WS18, incorporating the effect of grass conversion and reforestation treatments over time. Annual Q was computed on a May–April water year to minimize the effects of year-to-year changes in storage, as soils are generally at their wettest by the beginning of May. The empirical chronological-pairing model was fit using PROC NLIN (SAS v9.4, SAS Institute, Cary, NC) and had the following form: Q Ì́Ć_T=a+bQ_R+eM1t1+[M2c(h-1/(1+ãĂŰexpãĂŮˆ(-t2) ))] (1) where, Q Ì́Ć_T = predicted Q from treated watershed WS6 (mm yr-1), QR = measured Q from reference watershed WS18 (mm yr-1), M1 = management representing grass conversion; M1 = 1 for water years between and including 1960 and 1966, M1 = 0 otherwise, t1 = time since grass fertilization; t1 = water year – fertilization year for water years between and including 1960 and 1966 where fertilization years include water years 1959, 1961, and 1966, t1 = 0 otherwise, M2 = management representing reforestation after grass conversion; M2 = 1 for water years greater than or equal to 1967, M2 = 0 otherwise, t2 = time since reforestation after grass conversion; t2 = water year – 1967 for water years greater than or equal to 1967, t2 = 0 otherwise, P = annual precipitation (mm yr-1) a, b, c, e, h are fitted parameters. This overall modeling approach has been used in prior studies to assess the impact of forest management on Q (Ford et al., 2011; Kelly et al, 2016). The a+bQ_R term in EQ1 reflects the relationship between reference and treatment watersheds assuming no treatment. The increasing linear Q response (eM1t1 term in EQ1) accounts for the decline in annual grass production and water use after fertilization as noted by Hibbert (1969). The M2c(h-1/(1+ãĂŰexpãĂŮˆ(-t2) )) term in EQ1 accounts for the exponential decline in Q as the forest regenerates that has been observed in numerous paired watershed experiments (Swank et al., 1988). As in Ford et al. (2011), we define the Q treatment

response, DQ, as the difference between the observed Q in the treated watershed (QT) and that predicted by the model assuming no treatments had taken place (Q ÌĆ_T) : D_Q=Q_T-(Q ÌĆ_T; M1,M2=0). (2) The proportion of the variability explained by the model was quantified using the ratios of the error-to-total sum of squares and the total-to-error degrees of freedom as: R_adjusted^2=1-ãĂŰSSãĂŮ_E/ãĂŰSSãĂŮ_T × ãĂŰdfãĂŮ_T/ãĂŰdfãĂŮ_E . (3) Parameter estimates were interpreted as statistically significant at ïĄą = 0.05. Observed annual ET was computed as precipitation (P) – QT while expected ET with no treatment was computed as P - Q ÌĆ_T, both assuming the largely impermeable bedrock underlying the Basin that results in negligible deep groundwater losses (Douglass and Swank, 1972). Watershed P was estimated using a nearby eight inch (20.3 cm) National Weather Service standard rain gauge, SRG 96 (Laseter et al., 2012). The ET treatment response, DET, is then: D_ET=[ãĂŰP-QãĂŮ_T ]-([ãĂŰP-Q ÌĆãĂŮ_T ]; M1,M2=0) (4)."

The authors raise the very interesting point about non-stationary "controls" during these long-term paired watershed studies. It is probably the weakest point of this approach once they get past four or five decades (but when these were put in, who envisaged them going that long?). The difficulty is that I am not sure what one might do on this matter. Perhaps the authors could talk about this a bit more?

Response: Long term studies such as this help researchers understand the changes in "reference" watersheds as well as treated watersheds, "non-stationary" aspects. Despite non-stationarity in reference watersheds over the long term, the paired watershed approach remains our best tool for evaluating the effects of forest management on Q and ET. However, our approach of scaling up daily water use from the tree to the watershed has the potential to provide additional information regarding how both treatment and reference watersheds change over time.

So overall, it is a fine paper but a bit hard-going in its methods. The "discussion" probably needs a bit of tightening since it is to some extent speculative. I think that it would be worthy of a longer paper in which the methods are teased out and there is

more explanatory hydrograph detail. As an aside, I have always found it annoying that Dunford and Fletcher (1941) commented on the loss of the diurnal variation (if I recall correctly) but that no one ever seems to have looked at this again (i.e. how long did it take to reappear), and wonder if such a longer paper might also include indicators like this.

Response: we have revised the methods particularly with regard to the hydrology detail. See responses to Reviewer #1 above.

Please also note the supplement to this comment:
http://www.hydrol-earth-syst-sci-discuss.net/hess-2016-548/hess-2016-548-AC1-supplement.pdf

———————————————

[Figure]

**Legend**

‖‖‖‖‖‖ Coweeta Basin

▭ Watersheds

▨ Vegetation plots [C18]

SRG

WS18

[Figure]

[Figure]

Fig. 4. Changes in water yield

[Figure]

Growing season daily water use (kg day$^{-1}$)

Diffuse
Ring
Semi-ring
Evergreen shrub
Tracheid

DBH (cm)

**Fig. 6.** Growing season dail water use of tree species by xylem

[Figure]

Hydrol. Earth Syst. Sci. Discuss.,
doi:10.5194/hess-2016-548-RC1, 2016
Summary: This paper investigates the potential effects of land use and land cover changes on water yield (Q) and evapotranspiration (ET) by focusing on shifts in tree species composition during old-field succession. From a long data set (about 80 years) the authors observed a management induced change in vegetation from forest dominated by Quercus and Carya to grass and finally to regrown forest dominated by Liriodendron and Acer. These shifts were evident in the Q data. The conversion of forest to grass resulted in increases in Q, similar to previous studies that have studied the effects of clear-cutting on Q. The regrowth of forest, however, resulted in a decrease in Q and the shift in tree species composition resulted in Q becoming lower than in the original forest. The authors claim that this shift in Q was a result of changes in ET because of differences in water use among tree species. Liriodendron and Acer have a higher water use than Quercus and Carya. The authors also observed monthly changes in Q, especially in wetter months.

[Figure]

Contributions: Knowledge about how vegetation influences ET and Q is still not well

Hydrol. Earth Syst. Sci. Discuss.,
doi:10.5194/hess-2016-548-RC2, 2016

[Figure]

***Interactive comment on** "Water yield following forest–grass–
forest transitions"* **by Katherine J. Elliott et al.**

**Anonymous Referee #2**

Overall this is a good paper from a distinguished team using long-term data sets with a
history of quality measurement. That's the good part. The bad part is that the paper does
take a lot of following and reading, and after going through it a number of times, I'm still
not entirely sure about the methodology. I think that this may reflect on the reviewer
more than the writers, but the paper is a bit uncompromising in its terse pre- sentation of
information; to my mind that detracts a little from what is overall, a fine piece of work.
Some of my comments relate to the need to perhaps "help" the readers a little.
I think that the paper would be improved a little by better graphics. Firstly a picture or
two of WS6 at various stages would help. Similarly, a "time-line" of its treatments would
also be useful. A small map showing the various watersheds would be good, too.
**Response**: We have included a Figure with the map of the watershed locations, weirs,
and rain gages, now Figure 1. A time frame of WS6 treatments is provided in Table S1,
supporting information. On line 153 "The disturbance regime in WS6, the treated
watershed, was extensive (Table S1)."

I presume that the authors are trying to suppress a certain amount of detail – such as
the development of Equation 1 (which presumably goes back a long way). For the non-
hydrologists such an equation would be pretty enigmatic; I guess it is a judgement call for
the authors, but it is asking readers a lot to swallow this at one gulp, so to speak. Ditto the
frequency-pairing method.
**Response**: We have retained all of the Equations in the paper because they are important
in understanding the hydrology methodology. The methods are also well referenced. We
revised the section on methods based on comments from Reviewer #1 to provide
addition detail**:** to clarify the model and provide previous use (references) we revised text
on lines 222-261 as follows:

[revised manuscript text omitted]

Change in evapotranspiration ($D_{ET}$, calculated as $P - \hat{Q}_T$; where, $\boldsymbol{D_{ET}} = [P - Q_T] -$ $([P - \hat{Q}_T]; M1, M2 = 0)$) for the treated WS6 over time (bars). Solid lines are the standard errors of the mean prediction. We used the paired-watershed approach with WS18 as the reference. The year of harvest, conversion to Kentucky-31 fescue grass (*Festuca octiflora*) cover, fertilize, herbicide, and abandonment to allow forest regeneration are denoted by dashed lines.

---

## Author Comment (AC4) · 6 Jan 2017

Elliott et al. Anonymous Referee #2

Overall this is a good paper from a distinguished team using long-term data sets with a history of quality measurement. That's the good part. The bad part is that the paper does take a lot of following and reading, and after going through it a number of times, I'm still not entirely sure about the methodology. I think that this may reflect on the reviewer more than the writers, but the paper is a bit uncompromising in its terse presentation of information; to my mind that detracts a little from what is overall, a fine piece of work. Some of my comments relate to the need to perhaps "help" the readers

a little. I think that the paper would be improved a little by better graphics. Firstly a picture or two of WS6 at various stages would help. Similarly, a "time-line" of its treatments would also be useful. A small map showing the various watersheds would be good, too.

Response: We have included a Figure with the map of the watershed locations, weirs, and rain gages, now Figure 1. A time frame of WS6 treatments is provided in Table S1, supporting information. On line 153 "The disturbance regime in WS6, the treated watershed, was extensive (Table S1)."

I presume that the authors are trying to suppress a certain amount of detail – such as the development of Equation 1 (which presumably goes back a long way). For the non-hydrologists such an equation would be pretty enigmatic; I guess it is a judgement call for the authors, but it is asking readers a lot to swallow this at one gulp, so to speak. Ditto the frequency-pairing method.

Response: We have retained all of the Equations in the paper because they are important in understanding the hydrology methodology. The methods are also well referenced. We revised the section on methods based on comments from Reviewer #1 to provide addition detail: to clarify the model and provide previous use (references) we revised text on lines 222-261 as follows:

"We modeled WS6 annual Q and ET as a function of WS18, incorporating the effect of grass conversion and reforestation treatments over time. Annual Q was computed on a May–April water year to minimize the effects of year-to-year changes in storage, as soils are generally at their wettest by the beginning of May. The empirical chronological-pairing model was fit using PROC NLIN (SAS v9.4, SAS Institute, Cary, NC) and had the following form: Q Ì́C_T=a+bQ_R+eM1t1+[M2c(h-1/(1+ãĂŰexpãĂŮˆ(-t2) ))] (1) where, Q Ì́C_T = predicted Q from treated watershed WS6 (mm yr-1), QR = measured Q from reference watershed WS18 (mm yr-1), M1 = management representing grass conversion; M1 = 1 for water years between and including 1960 and

1966, M1 = 0 otherwise, t1 = time since grass fertilization; t1 = water year – fertilization year for water years between and including 1960 and 1966 where fertilization years include water years 1959, 1961, and 1966, t1 = 0 otherwise, M2 = management representing reforestation after grass conversion; M2 = 1 for water years greater than or equal to 1967, M2 = 0 otherwise, t2 = time since reforestation after grass conversion; t2 = water year – 1967 for water years greater than or equal to 1967, t2 = 0 otherwise, P = annual precipitation (mm yr-1) a, b, c, e, h are fitted parameters. This overall modeling approach has been used in prior studies to assess the impact of forest management on Q (Ford et al., 2011; Kelly et al, 2016). The a+bQ_R term in EQ1 reflects the relationship between reference and treatment watersheds assuming no treatment. The increasing linear Q response (eM1t1 term in EQ1) accounts for the decline in annual grass production and water use after fertilization as noted by Hibbert (1969). The M2c(h-1/(1+ãĂŰexpãĂŮˆ(-t2) )) term in EQ1 accounts for the exponential decline in Q as the forest regenerates that has been observed in numerous paired watershed experiments (Swank et al., 1988). As in Ford et al. (2011), we define the Q treatment response, DQ, as the difference between the observed Q in the treated watershed (QT) and that predicted by the model assuming no treatments had taken place (Q ÌĆ_T) : D_Q=Q_T-(Q ÌĆ_T; M1,M2=0). (2) The proportion of the variability explained by the model was quantified using the ratios of the error-to-total sum of squares and the total-to-error degrees of freedom as: R_adjustedˆ2=1-ãĂŰSSãĂŮ_E/ãĂŰSSãĂŮ_T × ãĂŰdfãĂŮ_T/ãĂŰdfãĂŮ_E . (3) Parameter estimates were interpreted as statistically significant at ïĄą = 0.05. Observed annual ET was computed as precipitation (P) – QT while expected ET with no treatment was computed as P - Q ÌĆ_T, both assuming the largely impermeable bedrock underlying the Basin that results in negligible deep groundwater losses (Douglass and Swank, 1972). Watershed P was estimated using a nearby eight inch (20.3 cm) National Weather Service standard rain gauge, SRG 96 (Laseter et al., 2012). The ET treatment response, DET, is then: D_ET=[ãĂŰP-QãĂŮ_T ]-([ãĂŰP-Q ÌĆãĂŮ_T ]; M1,M2=0) (4)."

The authors raise the very interesting point about non-stationary "controls" during these

long-term paired watershed studies. It is probably the weakest point of this approach once they get past four or five decades (but when these were put in, who envisaged them going that long?). The difficulty is that I am not sure what one might do on this matter. Perhaps the authors could talk about this a bit more?

Response: Long term studies such as this help researchers understand the changes in "reference" watersheds as well as treated watersheds, "non-stationary" aspects. Despite non-stationarity in reference watersheds over the long term, the paired watershed approach remains our best tool for evaluating the effects of forest management on Q and ET. However, our approach of scaling up daily water use from the tree to the watershed has the potential to provide additional information regarding how both treatment and reference watersheds change over time.

So overall, it is a fine paper but a bit hard-going in its methods. The "discussion" probably needs a bit of tightening since it is to some extent speculative. I think that it would be worthy of a longer paper in which the methods are teased out and there is more explanatory hydrograph detail. As an aside, I have always found it annoying that Dunford and Fletcher (1941) commented on the loss of the diurnal variation (if I recall correctly) but that no one ever seems to have looked at this again (i.e. how long did it take to reappear), and wonder if such a longer paper might also include indicators like this.

Response: we have revised the methods particularly with regard to the hydrology detail. See responses to Reviewer #1 above.

Please also note the supplement to this comment:
http://www.hydrol-earth-syst-sci-discuss.net/hess-2016-548/hess-2016-548-AC4-supplement.pdf

[Figure]

**Legend**

- ┈┈┈ Coweeta Basin
- ▭ Watersheds
- ▭ Vegetation plots
- **✚** Rain gauge
- ▼ Weir

**Elevation (m)**

High : 1590

Low : 672

N

WS6

SRG41

WS18

SRG96

WS14

0   0.25   0.5        1 Kilometers

**Fig. 1.** Map of the Coweeta Basin
Interactive
comment

[Figure]

**Fig. 2.** Mean (a) aboveground biomass, (b) leaf biomass

[Figure]

[Figure]

**Fig. 3.** Percetn aboveground biomass for the xylem

**Fig. 4.** Changes in water yield

**Fig. 5.** Changes in the cumulative distribution function

Growing season daily water use (kg day⁻¹) vs DBH (cm)

Legend:
- Diffuse (blue)
- Ring (dark red)
- Semi-ring (yellow)
- Evergreen shrub (green)
- Tracheid (gray)

**Fig. 6.** Growing season dail water use of tree species

[Figure]

**Fig. 7.** (a) Mean growing season daily water use (DWU)